# GWAS of three molecular traits highlights core genes and pathways alongside a highly polygenic background

Nasa Sinnott-Armstrong[1†]*, Sahin Naqvi[1,2†], Manuel Rivas[3], Jonathan K Pritchard[1,4]*

[1]Department of Genetics, Stanford University, Stanford, United States; [2]Department of Chemical and Systems Biology, Stanford University, Stanford, United States; [3]Department of Biomedical Data Sciences, Stanford University, Stanford, United States; [4]Department of Biology, Stanford University, Stanford, United States

**Abstract** Genome-wide association studies (GWAS) have been used to study the genetic basis of a wide variety of complex diseases and other traits. We describe UK Biobank GWAS results for three molecular traits—urate, IGF-1, and testosterone—with better-understood biology than most other complex traits. We find that many of the most significant hits are readily interpretable. We observe huge enrichment of associations near genes involved in the relevant biosynthesis, transport, or signaling pathways. We show how GWAS data illuminate the biology of each trait, including differences in testosterone regulation between females and males. At the same time, even these molecular traits are highly polygenic, with many thousands of variants spread across the genome contributing to trait variance. In summary, for these three molecular traits we identify strong enrichment of signal in putative core gene sets, even while most of the SNP-based heritability is driven by a massively polygenic background.

*For correspondence:
nasa@stanford.edu (NS-A);
pritch@stanford.edu (JKP)

[†]These authors contributed equally to this work

Competing interests: The authors declare that no competing interests exist.

## Introduction

One of the central goals of genetics is to understand how genetic variation, environmental factors, and other sources of variation, map into phenotypes. Understanding the mapping from genotype to phenotype is at the heart of fields as diverse as medical genetics, evolutionary biology, behavioral genetics, and plant and animal breeding.

The nature of the genotype-to-phenotype mapping has been a key motivating question ever since the start of modern genetics in the early 1900s. In those early days, the genetic basis of phenotypic variation was debated between the Mendelians, who were interested in discrete monogenic phenotypes, and the biometricians, who believed that Mendelian genetics were incompatible with the continuous distributions observed for height and many other traits. Those battles were largely resolved by Fisher's 1918 paper showing that a large number of Mendelian loci, each with proportionally weak effects, can approximate a continuous trait (*Fisher, 1918*; *Provine, 2001*; *Barton et al., 2017*; *Visscher and Goddard, 2019*). Taken to its extreme, this type of model is referred to as the 'infinitesimal model', and it laid the foundations for the growth of quantitative genetics in the 20th century (*Lynch and Walsh, 1998*).

Despite the importance of the infinitesimal model in the development of the field, for a long time this was mainly a theoretical abstraction. Even though some authors predicted early on that certain human diseases might be polygenic (*Penrose, 1953*; *Gottesman and Shields, 1967*), it was recognized that even a few loci (<10) can approximate infinitesimal predictions (*Thoday and Thompson, 1976*; *McGuffin and Huckle, 1990*). Thus, prior to the GWAS era it was entirely unclear how many loci would actually affect complex traits in practice (*Risch et al., 1999*; *Visscher and Goddard,*

*2019*). For example, in a 1989 review of quantitative genetics, Barton and Turelli wrote that 'we still do not know whether the number of loci responsible for most genetic variation is small (5–20) or large (100 or more)' (*Barton and Turelli, 1989*). Consistent with this, practioners of human genetics in the pre-GWAS era expected that we might be looking for a small handful of genes per trait; in the 1990s, this motivated hundreds of small studies of complex traits that were only powered to detect large-effect loci. In one typical example, Risch and Merikangas' foundational 1996 paper on association mapping computed the power for common variants with relative risks between the alternate homozygotes ranging from 2.25 to 16 (*Risch and Merikangas, 1996*): effect sizes that we now know were unrealistically high.

The advent of GWAS, starting around fifteen years ago, completely transformed our understanding of the genetic basis of a wide variety of human complex traits and diseases (*Claussnitzer et al., 2020*). While early GWAS studies showed the power of this approach to identify significant and replicable signals, it quickly became clear that the lead variants generally explain only small fractions of the heritability of the corresponding traits (*Weedon et al., 2008*; *Goldstein, 2009*). The limited explanatory power of the detected loci became known as the ''mystery of missing heritability'' (*Manolio et al., 2009*): a mystery that was largely resolved by work showing that most of the heritability is due to the presence of many sub-significant causal variants (*Purcell et al., 2009*; *Yang et al., 2010*). Subsequent work has shown that for most traits the bulk of the SNP-based heritability is spread widely and surprisingly uniformly across the genome (*Loh et al., 2015*; *Shi et al., 2016*; *O'Connor, 2020*), and that most complex traits are in fact *enormously* polygenic, with various studies estimating >10,000 or even >100,000 causal variants per trait (*Zhang et al., 2018*; *Frei et al., 2019*; *O'Connor et al., 2019*).

## Why are complex traits so polygenic?

These findings raise a *mechanistic* question of how to understand the biological processes that link genotype to phenotype. How should we understand the observations that the lead variants for a typical trait contribute only a small fraction of the heritability, while most of the heritability is driven by tens of thousands of variants, presumably mediated through thousands of genes?

One hypothesis for polygenicity is the observation that many disease or behavioral endpoints are impacted by multiple distinct processes, or endophenotypes (*Turkheimer, 2000*; *Gottesman and Gould, 2003*): for example, diabetes is affected by lipids, adiposity and β-cell function (*Udler, 2019*). For such traits, the genetic basis of the endpoint phenotype is expected to reflect the genetics of all the intermediate phenotypes. While this is surely true, it seems unlikely to fully resolve the question of why specific phenotypes can be affected by tens of thousands of variants, unless the endophenotypes themselves are highly polygenic. Indeed, as we will show in this paper, even relatively 'simple' molecular traits such as urate levels can be hugely polygenic, implying that we need additional explanations for high polygenicity.

In recent work, we proposed an alternative model for understanding extreme polygenicity, namely that it may be a consequence of the architecture of gene regulatory networks (*Boyle et al., 2017*; *Liu et al., 2019*). Work from several groups has shown that, for an average gene, most of the heritability in gene expression results from large numbers of small trans effects (*Price et al., 2008*; *Liu et al., 2019*). Building on this observation, we proposed a conceptual model in which there is a set of 'core' genes, defined as genes with a direct effect on the trait that is not mediated through regulation of other genes. Meanwhile, other genes that are expressed in trait-relevant cell types are referred to as 'peripheral' genes, and can matter if they affect the expression of core genes. By this definition, transcription factors (TFs) are considered peripheral, but we refer to TFs with coordinated effects on multiple core genes as 'master regulators' to acknowledge their special roles. This model primarily denotes different categories of *genes* (rather than variants) with respect to their roles in trait variation; we discuss how these distinctions may apply to variant effect sizes later in the text.

We proposed that variants near core genes contribute only a small fraction of the heritability, and that instead most trait variance is due to huge numbers of trans-regulatory effects from SNPs with cis-effects on peripheral genes. In what we referred to as the 'omnigenic' extreme, potentially any gene expressed in trait-relevant cell types could affect the trait through effects on core gene expression. (Note that this does not mean that every gene would in fact have associated variants, as presumably the distribution of peripheral gene effect sizes would be centered on zero, and in practice not all genes have regulatory variants).

While the omnigenic model is broadly consistent with observations on cis and trans heritability of expression (*Liu et al., 2019*), it has been difficult to evaluate the model in detail because for most diseases and other traits we know little in advance about which genes are likely to be directly involved in disease biology. Recent efforts to systematically nominate core genes have primarily relied upon associations identified in rare, monogenic disorders (*Vuckovic et al., 2020*); while promising, such approaches are inherently limited by the ability to discover rare gene-disease associations, which can depend upon a number of factors. Furthermore, we still have highly incomplete information about cellular regulatory networks and trans-eQTLs.

Here, we focus on three molecular traits that are unusually tractable in order to gain insights into the roles of core genes. This work illustrates two key parts of the model: (1) the existence and identity of sets of core genes for each trait and (2) that the core genes contribute only a small fraction of the heritability. We do not directly assess the role of trans-regulatory networks for these traits as well-powered trans-eQTL data do not exist for the relevant cell types.

## GWAS of model traits: three vignettes

We investigate the genetic architecture underlying variation in three molecular traits: serum urate, IGF-1, and testosterone levels. For each of these traits, we know a great deal in advance about the key organs, biological processes and genes that might control these traits.

This stands in contrast to many of the traits that have been studied extensively with GWAS, such as schizophrenia (*Ripke et al., 2014*; *Ripke et al., 2020*; which is poorly understood at the molecular level) or height (*Wood et al., 2014*; where we understand more of the underlying biology, but for which a large number of different biological processes contribute variance). We do now know various examples of core genes or master regulators for specific traits (e.g. *Sekar et al., 2016*; *Small et al., 2011*; *Small et al., 2018*), but there are few traits where we understand the roles of more than a few of the lead genes. Among the clearest examples in which a whole suite of core genes have been identified are for plasma lipid levels (e.g. *Liu et al., 2017*; *Lu et al., 2017*; *Hoffmann et al., 2018*, reviewed by *Dron and Hegele, 2016*; *Liu et al., 2019*); and for inflammatory bowel disease (*de Lange et al., 2017*).

As described in more detail below, we performed GWAS for each of these traits in around 300,000 white British individuals from the UK Biobank (*Bycroft et al., 2018*). For all three traits many of the most significant hits are highly interpretable–a marked difference from GWAS of typical disease traits. While these three molecular traits highlight different types of lead genes and molecular processes, they also have strikingly similar overall architectures: the top hits are generally close to genes with known biological relevance to the trait in question, and all three traits show strong enrichment in relevant gene sets. Most of these genes would be considered core genes (or occasionally master regulators) in the sense of *Liu et al., 2019*.

At the same time, however, variants near the lead genes and pathways explain only a modest fraction of the heritability. Aside from one major-effect variant for urate, the lead pathways explain ~10% of the SNP-based heritability. Instead, most of the SNP-based heritability is due to a highly polygenic background, which we conservatively estimate as being due to around 10,000 causal variants per trait.

In summary, these three molecular traits provide points of both contrast and similarity to the architectures of disease phenotypes. From one point of view they are clearly simpler, successfully identifying known biological processes to an extent that is highly unusual for disease GWAS. At the same time, the most significant hits sit on a hugely polygenic background that is reminiscent of GWAS for more-complex traits.

## Results

Our analyses make use of GWAS results that we reported previously on blood and urine biomarkers (*Sinnott-Armstrong et al., 2021*), with minor modifications. In the present paper, we report four primary GWAS analyses: urate, IGF-1, and testosterone in females and males separately. Prior to each GWAS, we adjusted the phenotypes by regressing the measured phenotypes against age, sex (urate and IGF-1 only), self-reported ethnicity, the top 40 principal components of genotype, assessment center and month of assessment, sample dilution and processing batch, as well as relevant pairwise interactions of these variables (Materials and methods).

We then performed GWAS on the phenotype residuals in White British participants. For the GWAS we used variants imputed using the Haplotype Reference Consortium with MAF >0.1% and INFO >0.3 (Materials and methods), yielding a total of 16M variants. The final sample sizes were 318,526 for urate, 317,114 for IGF-1, 142,778 for female testosterone, and 146,339 for male testosterone. One important goal of our paper is to identify the genes and pathways that contribute most to variation in each trait. For gene set-enrichment analyses, we annotated gene sets using a combination of KEGG (*Kanehisa and Goto, 2000*) and previous trait-specific reviews, as noted in the text. We considered a gene to be 'close' to a genome-wide significant signal if it was within 100 kb of at least one lead SNP with p<5e-8. The annotations of lead signals on the Manhattan plots were generally guided by identifying nearby genes within the above-described enriched gene sets, or occasionally other strong nearby candidates.

## Genetics of serum urate levels

Urate is a small molecule ($C_5H_4N_4O_3$) that arises as a metabolic by-product of purine metabolism and is released into the blood serum. Serum urate levels are regulated by the kidneys, where a set of transporters shuttle urate between the blood and urine; excess urate is excreted via urine. Urate is used as a clinical biomarker due to its associations with several diseases. Excessively high levels of urate can result in the formation of needle-like crystals of urate in the joints, a condition known as gout. High urate levels are also linked to diabetes, cardiovascular disease, and kidney stones.

The genetics of urate have been examined previously by several groups (*Woodward et al., 2009*; *Köttgen et al., 2013*; *Nakayama et al., 2017*; *Nakatochi et al., 2019*; *Boocock et al., 2019*; *Tin et al., 2019* and recently reviewed by *Major et al., 2018*). The three strongest signals for urate lie in solute carrier genes: SLC2A9, ABCG2, and SLC22A11/SLC22A12. A recent trans-ancestry analysis of 457 k individuals identified 183 genome-wide significant loci (*Tin et al., 2019*); their primary analysis did not include UK Biobank. Among other results, this study highlighted genetic correlations of urate with gout and various metabolic traits; tissue enrichment signals in kidney and liver; and genetic signals at the master regulators for kidney and liver development HNF1A and HNF4A.

Performing GWAS of urate in the UK Biobank data set, we identified 222 independent genome-wide significant signals, summarized in *Figure 1A* (further details in *Supplementary file 1*). Remarkably, six of the 10 most significant signals are located within 100 kb of a urate solute transport gene. A recent review identified 10 genes that are involved in urate solute transport in the kidneys (*Wright et al., 2010*; *Anzai et al., 2007*); in addition to the six transporters with extremely strong signals, two additional transporters have weaker, yet still genome-wide significant signals (*Figure 1B*). Hence, GWAS highlights eight out of 10 annotated urate transporters, although some transporters were originally identified using early GWAS for urate levels. The two genes in the pathway that do not have hits (SMCT1 and SMCT2; also known as SLC5A8 and SLC5A12) do not directly transport urate, but instead transport monocarboxylate substrates for URAT1 to increase reabsorption rate (*Bobulescu and Moe, 2012*) and thus may be less direct regulators of urate levels.

Among the other top hits, five are close to transcription factors involved in kidney and liver development (HNF4G, HNF1A, HNF4A, HLF, and MAF). These are not part of a globally enriched gene set, but recent functional work has shown that the associated missense variant in HNF4A results in differential regulation of the urate solute carrier ABCG2 (*Tin et al., 2019*), while the MAF association has been shown to regulate SLC5A8 (*Leask et al., 2018*). Finally, two other loci show large signals: a missense variant in INHBC, a TGF-family hormone, and a variant in/near GCKR, a glucose-enzyme regulator. Both variants have highly pleiotropic effects on many biomarkers, although the mechanisms pertaining to urate levels are unclear.

While most of the top hits are likely associated with kidney function, we wanted to test whether other tissues contribute to the overall SNP-based heritability (*Figure 1C*). To this end, we used stratified LD Score regression to estimate the polygenic contribution of regulatory regions in 10 previously defined tissue groupings (*Finucane et al., 2015*). Serum urate SNP-based heritability was most-highly enriched in kidney regulatory regions (29-fold compared to the genome-wide average SNP, p=1.9e-13), while other cell types were enriched around 8-fold (*Figure 1—figure supplement 1*; see also *Tin et al., 2019*). We hypothesized that the enrichment for other tissues might be driven by elements shared between kidney and other cell types. Indeed, when we removed active kidney regions from the regulatory annotations for other tissues, this eliminated most of the signal found in

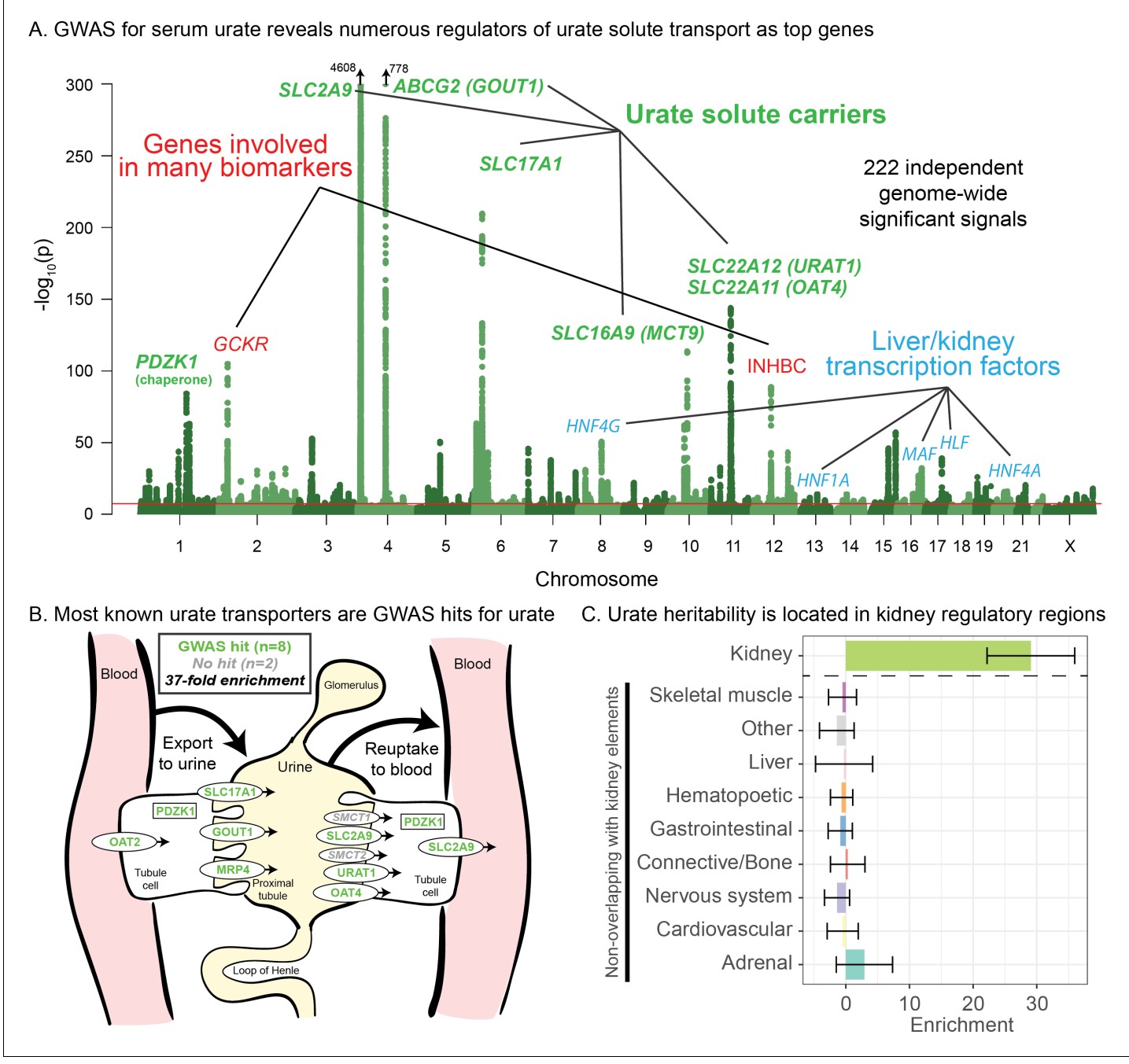

**Figure 1.** Genetic basis of serum urate variation. (**A**) Genome-wide associations with serum urate levels in the UK Biobank. Candidate genes that may drive the most significant signals are indicated; in most cases in the paper, the indicated genes are within 100 kb of the corresponding lead SNPs. (**B**) Eight out of 10 genes that were previously annotated as being involved in urate transport (**Wright et al., 2010**; **Anzai et al., 2007**) are within 100 kb of a genome-wide significant signal. The signal at MCT9 is excluded from figure and enrichment due to its uncertain position in the pathway (**Fisel et al., 2018**). (**C**) Urate SNP-based heritability is highly enriched in kidney regulatory regions compared to the genome-wide background (analysis using stratified LD Score regression). Other tissues show little or no enrichment after removing regions that are active in kidney. See **Figure 1—figure supplement 1** for the uncorrected analysis.

The online version of this article includes the following figure supplement(s) for figure 1:

**Figure supplement 1.** Estimates of serum urate SNP-based heritability within cell and tissue group annotations using LD Score regression (**Finucane et al., 2015**).

other cell types (*Figure 1C*). Thus, our analysis supports the inference that most serum urate heritability is driven by kidney regulatory variation.

Finally, while these signals emphasize the role of the kidneys in setting urate levels, we wanted to test specifically for a role of urate synthesis (similar to recent work on glycine [*Wittemans et al., 2019*]). The urate molecule is the final step of purine breakdown; most purines are present in tri- and monophosphates of adenosine and guanosine, where they act as signaling molecules, energy sources for cells, and nucleic acid precursors. The breakdown pathways are well known, including the genes that catalyze these steps (*Figure 2A*).

Overall, we found that genes in the urate metabolic pathway show a modest enrichment for GWAS hits relative to all annotated, protein coding genes as a background (2.1-fold, p=0.017; *Figure 2B*). XDH, which catalyzes the last step of urate synthesis, has an adjacent GWAS hit, as do a number of upstream regulators of urate synthesis. Nonetheless, the overall level of signal in the synthesis pathway is modest compared to that seen for kidney urate transporters, suggesting that synthesis, while it plays a role in common variation in urate levels, is secondary to the secretion pathway. In contrast, remarkably, nearly all of the kidney urate transporter genes are close to genomewide significant signals; there are additional strong signals in kidney transcription factors, as well as a strong polygenic background in kidney regulatory regions.

## Genetics of IGF-1 levels

Our second vignette considers the genetic basis of IGF-1 (insulin-like growth factor 1) levels. The IGF-1 protein is a key component of a signaling cascade that connects the release of growth hormone to anabolic effects on cell growth in peripheral tissues (*Laron, 2001*). Growth hormone is produced in the pituitary gland and circulated around the body; in the liver, growth hormone triggers the JAK-STAT pathway leading, among other things, to IGF-1 secretion. IGF-1 binding to IGF-1 receptor, in turn, activates the RAS and AKT signaling cascades in peripheral tissues. IGF-1 is used as a clinical biomarker of growth hormone levels and pituitary function, as it has substantially more stable levels and a longer half-life than growth hormone itself. The growth hormone–IGF axis is a conserved regulator of longevity in diverse invertebrates and possibly mammals (*van Heemst, 2010*). In humans, both low and high levels of IGF-1 have been associated with increased mortality from cancer and cardiovascular disease (*Burgers et al., 2011*). We note that while IGF-1 rises sharply in puberty, our analyses are focused on middle-aged individuals. IGF-1 is a major effect locus for

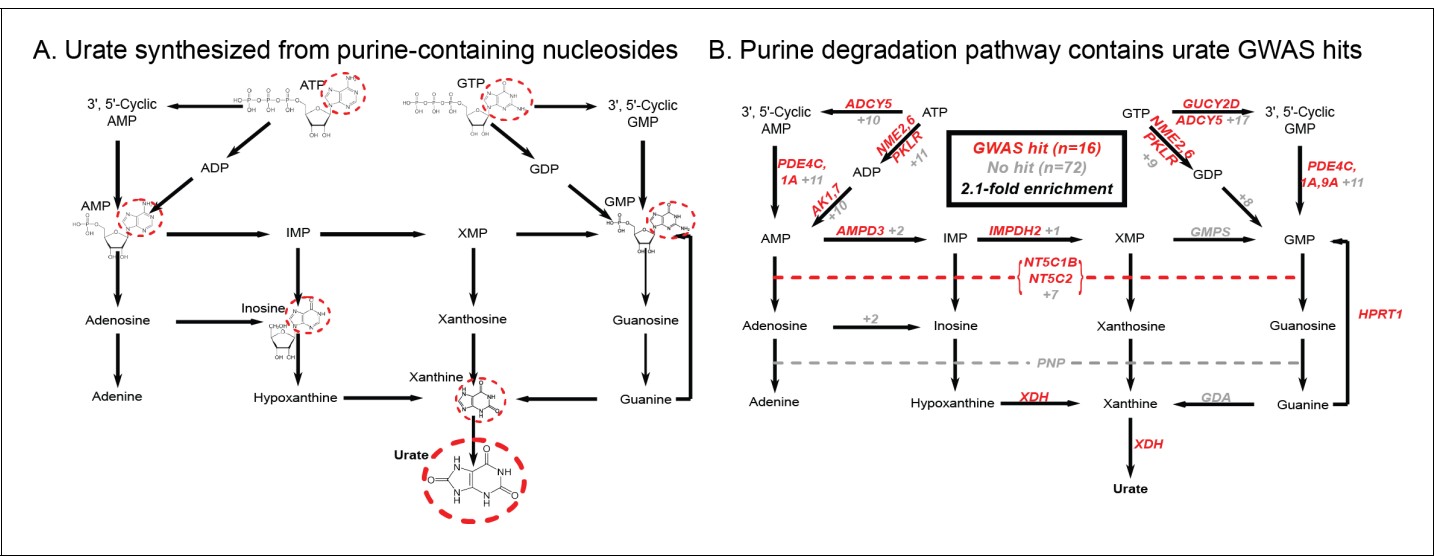

**Figure 2.** Modest enrichment of signals among genes involved in urate biosynthesis. (**A**) Urate is a byproduct of the purine biosynthesis pathway. The urate component of each molecule is highlighted. (**B**) The same pathway indicating genes that catalyze each step. Genes with a genome-wide significant signal within 100 kb are indicated in red; numbers in gray indicate the presence of additional genes without signals. Pathway adapted from KEGG.

body size in dogs (*Sutter et al., 2007*), and IGF-1 levels are positively associated with height in UK Biobank (*Figure 3—figure supplement 1*).

Previous GWAS for IGF-1, using up to 31,000 individuals, identified around half a dozen genome-wide significant loci (*Kaplan et al., 2011*; *Teumer et al., 2016*). The significant loci included IGF-1 itself and a signal close to its binding partner IGFBP3.

In our GWAS of serum IGF-1 levels in 317,000 unrelated White British individuals, we found a total of 354 distinct association signals at genome-wide significance (*Figure 3*, further details in *Supplementary file 2*). Eight of the most significant signals are key parts of the IGF-1 pathway (*Figure 4*). The top hit is an intergenic SNP between IGFBP3 and another gene, TNS3 (*Supplementary file 2*; p=1e-837). IGFBP3 encodes the main transport protein for IGF-1 and IGF-2 in the bloodstream (*Firth and Baxter, 2002*). The next most significant hits are at the IGF-1 locus itself and at its paralog IGF-2. Two other lead hits are associated with the IGF transport complex IGFBP: IGFALS, which is an IGFBP cofactor that also binds IGF-1 in serum (*Baxter et al., 1989*), and PAPPA2, a protease which cleaves and negatively regulates IGFBPs (*Overgaard et al., 2001*). Three other lead hits lie elsewhere in the growth hormone–IGF axis: GHSR is a pituitary-expressed receptor for the signaling protein ghrelin which negatively regulates the growth hormone (GH) signaling pathway upstream of IGF-1 (*Laron, 2001*); and FOXO3 and RIN2 lie in downstream signaling pathways (*Stitt et al., 2004*).

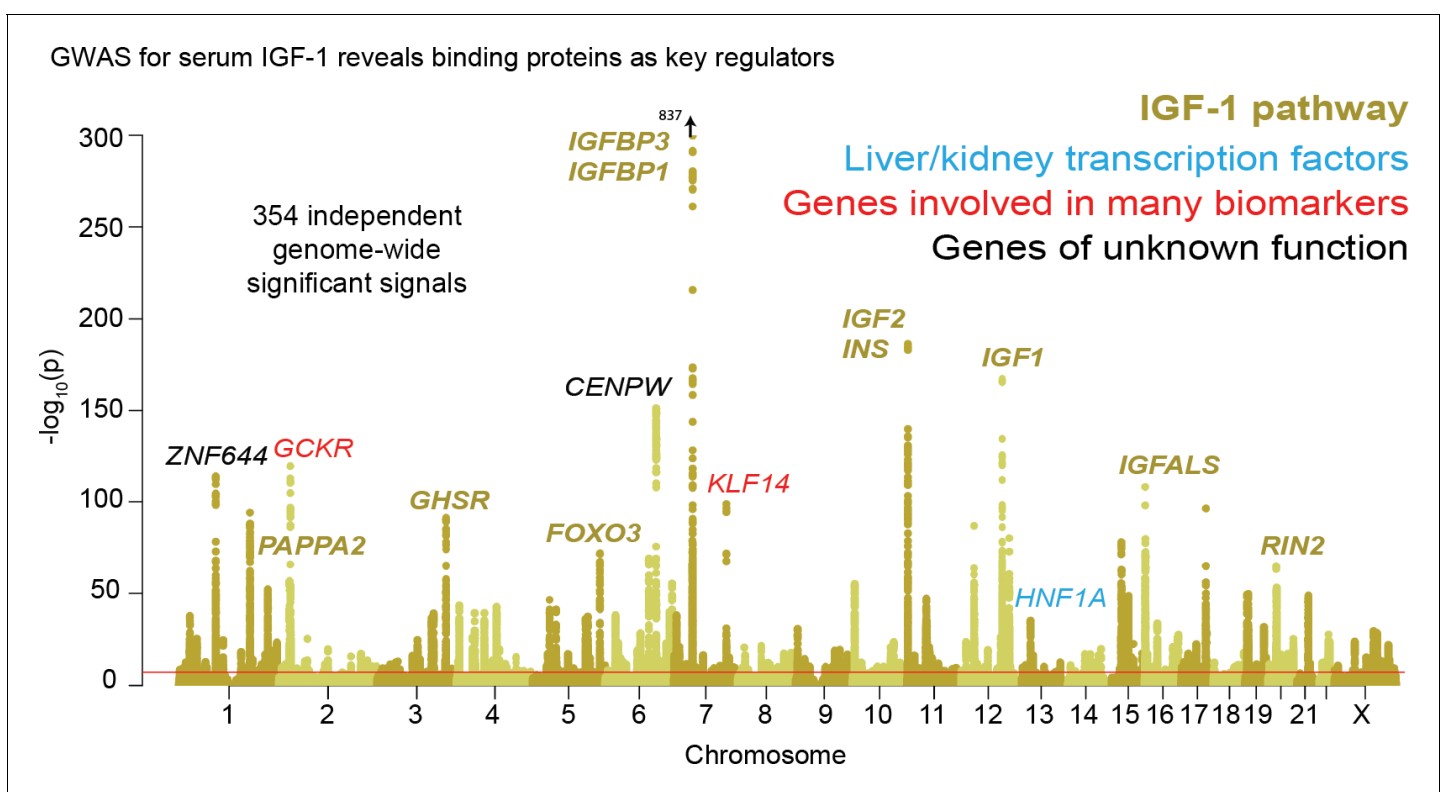

**Figure 3.** Genetic basis of IGF-1 variation. Manhattan plot showing the locations of major genes associated with IGF-1 levels in the IGF-1 pathway (yellow), transcription factor (blue), pleiotropic gene (red), or unknown function (black) genes sets.

The online version of this article includes the following figure supplement(s) for figure 3:

**Figure supplement 1.** Covariate-adjusted IGF-1 levels are significantly associated with covariate-adjusted height in UK Biobank.

**Figure supplement 2.** QQ-plot testing for epistasis plots all pairs of lead variants with $p < 1e - 20$ for IGF-1 levels (Materials and methods).

**Figure supplement 3.** QQ-plot testing for non-additivity at IGF-1 associated SNPs.

**Figure supplement 4.** A genome-wide association study for paired differences in effect size by SLC2A9 genotype.

**Figure supplement 5.** Non-additivity in serum urate concentrations at chr4:10107439 C>T, chr1:15816768 CACAT>C, chr4:89082319 T>A, chr4:22807237 A>G, and chr10:61469538 T>A; at chr2:25946813 C>T in IGF-1; and in female testosterone at chr19:10471462 C>T and male testosterone at chr1:107563243 G>T and chr17:7560835 T>G.

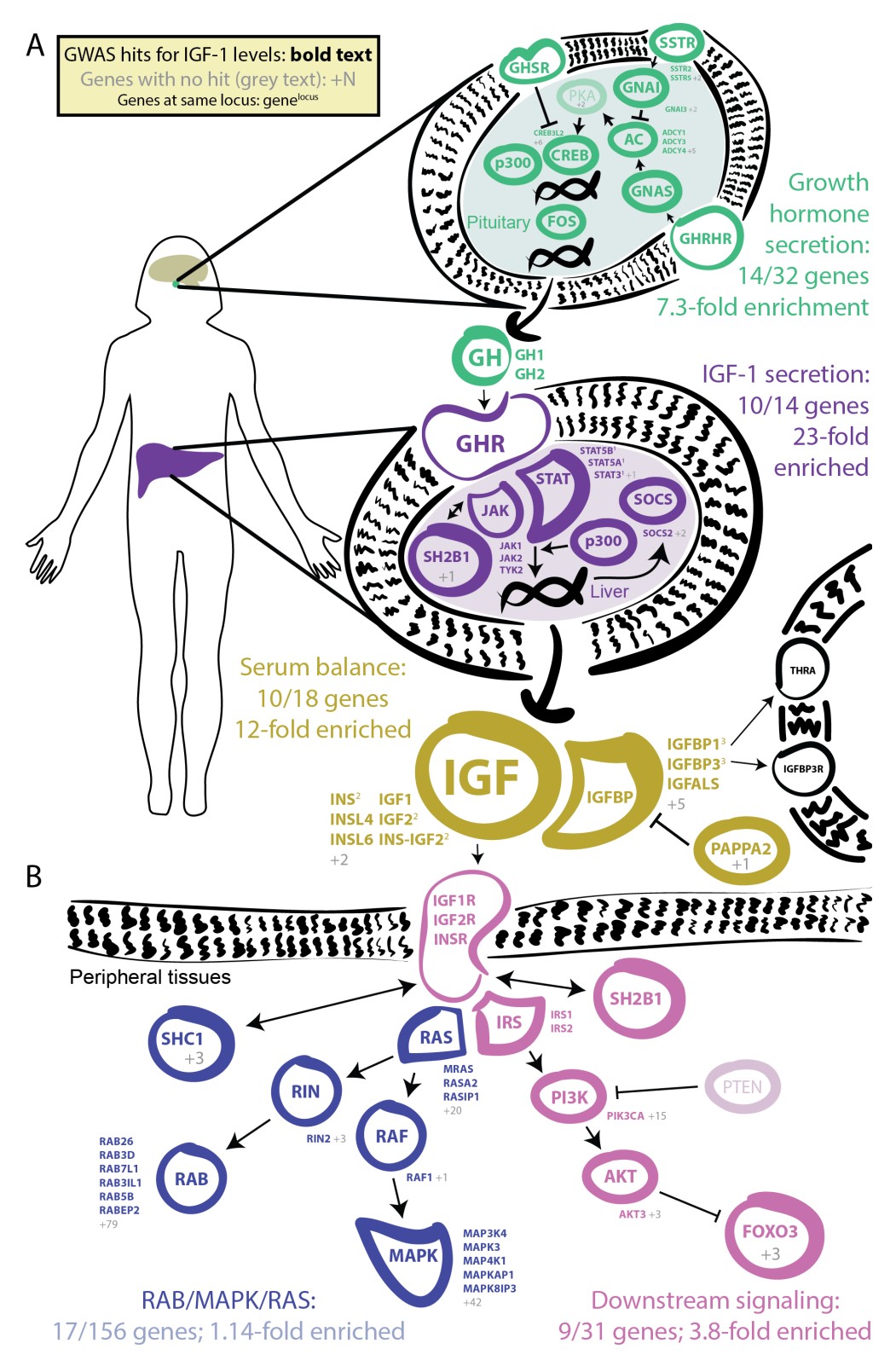

**Figure 4.** GWAS hits in the IGF-1 pathway. Bolded and colored gene names indicate that the gene is within 100 kb of a genome-wide signficant hit. Gray names indicate absence of a genome-wide signficant hit; gray numbers indicate that multiple genes in the same part of the pathway with no hit. Superscript numbers indicate that multiple genes are located within the same locus and hence may not have independent hits. (**A**) Upstream pathway that controls regulation of IGF-1 secretion into the bloodstream. (**B**) Downstream pathway that controls regulation of IGF-1 response.

Additional highly significant hits that are not directly involved in the growth hormone–IGF pathway include the liver transcription factor HNF1A (also associated with urate [*Tin et al., 2019*]); variants near two genes–GCKR and KLF14–that are involved in many biomarkers, although to our knowledge the mechanism is unclear; and variants at two additional genes CENPW and ZNF644.

Given the numerous lead signals in the IGF-1 signaling cascade, we sought to comprehensively annotate all GWAS hits within the cascade and its sub-pathways. We compiled lists of the genes from KEGG and relevant reviews from five major pathways in the growth hormone–IGF axis (*Figure 4*, Materials and methods). Four of the five pathways show extremely strong enrichment of GWAS signals. The first pathway regulates *growth hormone secretion*, acting in the pituitary to integrate ghrelin and growth hormone releasing hormone signals and produce growth hormone. This pathway shows strong enrichment, with 14 out of 32 genes within 100 kb of a genome-wide significant signal (7.3-fold enrichment, Fisher's exact p=5.4e-7). The second pathway, *IGF-1 secretion*, acts in the liver, where growth hormone triggers JAK-STAT signaling, leading to IGF-1 production and secretion (*Dehkhoda et al., 2018*). This pathway again shows very strong enrichment of GWAS signals (10/14 genes, 23-fold enrichment, p=4.9e-8). The third pathway, *serum balance of IGF*, relates to IGF-1 itself, and its paralogs, as well as other binding partners and their regulators in the serum. Here 10/18 genes have GWAS hits (11.7-fold enrichment, p=1.5e-6).

We also considered two downstream signaling pathways that transmit the IGF signal into peripheral tissues. Most notably, many of the genes in the *AKT branch of the IGF-1 signaling cascade* were close to a genome-wide significant association including FOXO3 (9/31 genes; 3.8-fold enrichment, p=0.002). In contrast, the *RAB/MAPK/RAS pathway* was not enriched overall (p=0.59), although one key signaling molecule (RIN2) in this pathway was located at one of the strongest hits genome-wide. The observation of strong signals downstream of IGF-1 suggests the presence of feedback loops contributing to IGF-1 regulation. This is consistent with work proposing negative feedback from downstream pathways including AKT and MAPK to growth hormone activity (*Li et al., 2009*).

Lastly, given that most of the strongest hits lie in the same pathway, we were curious whether there might be evidence for epistatic or non-additive interactions. Experiments in molecular and model organism biology regularly find interaction effects between genes that are close together in pathways (*Tong et al., 2004*; *Scanga et al., 2000*; *Bassik et al., 2013*; *Fischer et al., 2015*; *Wang et al., 2014*), but evidence for epistatic interactions between GWAS variants is extraordinarily rare (*Ritchie and Van Steen, 2018*), potentially due to GWAS hits lying in unrelated pathways, having modest effect sizes, or most often not representing the causal variant. We found no signal of epistasis among the 77 most significant (p<1e-20) lead SNPs for IGF-1 (*Figure 3—figure supplement 2*), and weak enrichment of signal among the top 38 urate lead SNPs (*Figure 3—figure supplement 2* inset; see Materials and methods). Similarly, IGF-1 lead SNPs (p<5e-8) showed weak, global inflation of test statistics for departures from additivity (e.g. dominance or recessivity) (*Figure 3—figure supplement 3*), while the two most significant urate hits showed significant minor dominant (SLC2A9) and minor recessive (ABCG2) effects that were nevertheless substantially smaller than the additive effects (*Figure 3—figure supplement 3* inset). Genome-wide paired difference tests for the SLC2A9 variant showed no signal (*Figure 3—figure supplement 4*). Building upon previous studies (*Wei et al., 2014*; *Zaitlen et al., 2013*) that have found little evidence of epistasis or dominance in human populations, these results indicate that non-linear genotype effects, while likely present to some degree, are substantially weaker than additive components, even when the strongest effects are concentrated in the same biological pathways and would thus be more likely to show epistasis.

In summary for IGF-1, we found 354 distinct associations that surpass genome-wide significance. The lead variants show strong enrichment across most components of the growth hormone-IGF axis, including the downstream AKT signaling arm, suggesting regulatory feedback. Among the strongest hits we also find involvement of one transcription factor (HNF1A) and two other genes of unclear functions (GCKR and KLF14) that have pleiotropic effects on multiple biomarkers, perhaps due to overall effects on liver and kidney development.

## Testosterone

Our third vignette describes the genetic basis of testosterone levels. Testosterone is a four carbon-ring molecule ($C_{19}H_{28}O_2$) that functions as an anabolic steroid and is the primary male sex hormone. Testosterone is crucial for the development of male reproductive organs and secondary sex

characteristics, while also having important functions in muscle mass and bone growth and density in both females and males (*Tracz et al., 2006*; *Herbst and Bhasin, 2004*). Circulating testosterone levels range from about 0.3 to 2 nmol/L in females and 8 to 33 nmol/L in males (*Figure 5—figure supplement 1*).

Testosterone is synthesized from cholesterol as one possible product of the steroid biosynthesis pathway. Synthesis occurs primarily in the testis in males, and in the ovary and adrenal glands in females. Testosterone production is stimulated by the hypothalmic-pituitary-gonadal (HPG) axis: gonadotropin-releasing hormone (GnRH) signals from the hypothalamus to the pituitary to cause production and secretion of luteinizing hormone (LH); LH in turn signals to the gonads to produce testosterone. The HPG axis is subject to a negative feedback loop as testosterone inhibits production of GnRH and LH by the hypothalamus and pituitary to ensure tight control of testosterone levels (*Javorsky et al., 2017*). Testosterone acts on target tissues via binding to the androgen receptor (AR) which in turn regulates downstream genes. Approximately half of the circulating testosterone (~40% in males, ~60% in females [*Dunn et al., 1981*]) is bound to sex hormone binding globulin (SHBG) and is generally considered non-bioavailable. Testosterone breakdown occurs primarily in the liver in both females and males.

Previous GWAS for serum testosterone levels studied up to 9000 males, together finding three genome-wide significant loci, the most significant of which was at the SHBG gene (*Ohlsson et al., 2011*; *Jin et al., 2012*). While this paper was in preparation, two studies reported large-scale GWAS of testosterone levels in UKBB individuals, finding significant sex-specific genetic effects (*Flynn et al., 2021*; *Ruth et al., 2020*). Previous studies of young adults found minimal correlation of salivary testosterone levels between opposite-sex dizygotic twins (*Grotzinger et al., 2018*). In our preliminary analysis, we found that testosterone shows minimal genetic correlation between the sexes, in contrast to other biomarkers including urate and IGF-1 (*Figure 7—figure supplement 1*). We therefore performed sex-stratified GWAS of testosterone, in contrast to the combined analysis used for urate and IGF-1.

Here, we performed testosterone GWAS in UKBB females (N = 142,778) and males (N = 146,339) separately. We discovered 79 and 127 independent genome-wide significant signals in females and males, respectively (*Figure 5*, further details in *Supplementary file 3–4*). We note that a recent paper reported larger numbers of independent genome-wide significant signals (245 and 231 in females and males, respectively); this was likely due to the inclusion of individuals with broader European ancestry, as well as a less stringent definition of independence used by Ruth et al (*Ruth et al., 2020*).

In females, six of the most significant signals are close to genes involved in testosterone biosythesis (*Figure 5A*); together these results suggest that the steroid biosynthesis pathway is the primary controller of female testosterone levels. Among these, the top hit is at a locus containing three genes involved in hydroxylation of testosterone and estrone, CYP3A4, CYP3A5, and CYP3A7 (*Kandel et al., 2017*; *Lee et al., 2003*; *Kuehl et al., 2001*). Two other lead hits (MCM9 and FGF9) are involved in gonad development (*Lutzmann et al., 2012*; *Wood-Trageser et al., 2014*; *Colvin et al., 2001*).

Strikingly, and in agreement with recent studies and in agreement with recent studies (*Flynn et al., 2021*; *Ruth et al., 2020*), the lead hits in males are largely non-overlapping with those from females. Overall, the male hits affect a larger number of distinct processes. Three of the most significant signals affect the steroid biosynthesis pathway (SRD5A2, UGT2B15, and AKR1C); three are involved in either upstream activation (NR0B2) (*Vega et al., 2015*) or downstream signaling (the androgen receptor, AR, and its co-chaperone FKBP4), respectively; and two have been implicated in the development of the GnRH-releasing function of the hypothalamus (KAL1) (*Franco et al., 1991*) or the gonads (NR2F2) (*Qin et al., 2008*). However, the largest category, including the most significant hit overall, is for a group of eight distinct variants previously shown to affect sex hormone binding globulin (SHBG) levels (*Coviello et al., 2012*). SHBG is one of the main binding partners for testosterone–we will discuss the significance of SHBG below.

## Steroid biosynthesis

Given our observation of numerous lead hits near steroid hormone biosynthesis genes, we curated the female and male hits in the KEGG pathway (*Figure 6*). We observed that nearly all major steps of the pathway contained a gene near a genome-wide significant SNP in either females or males: 31

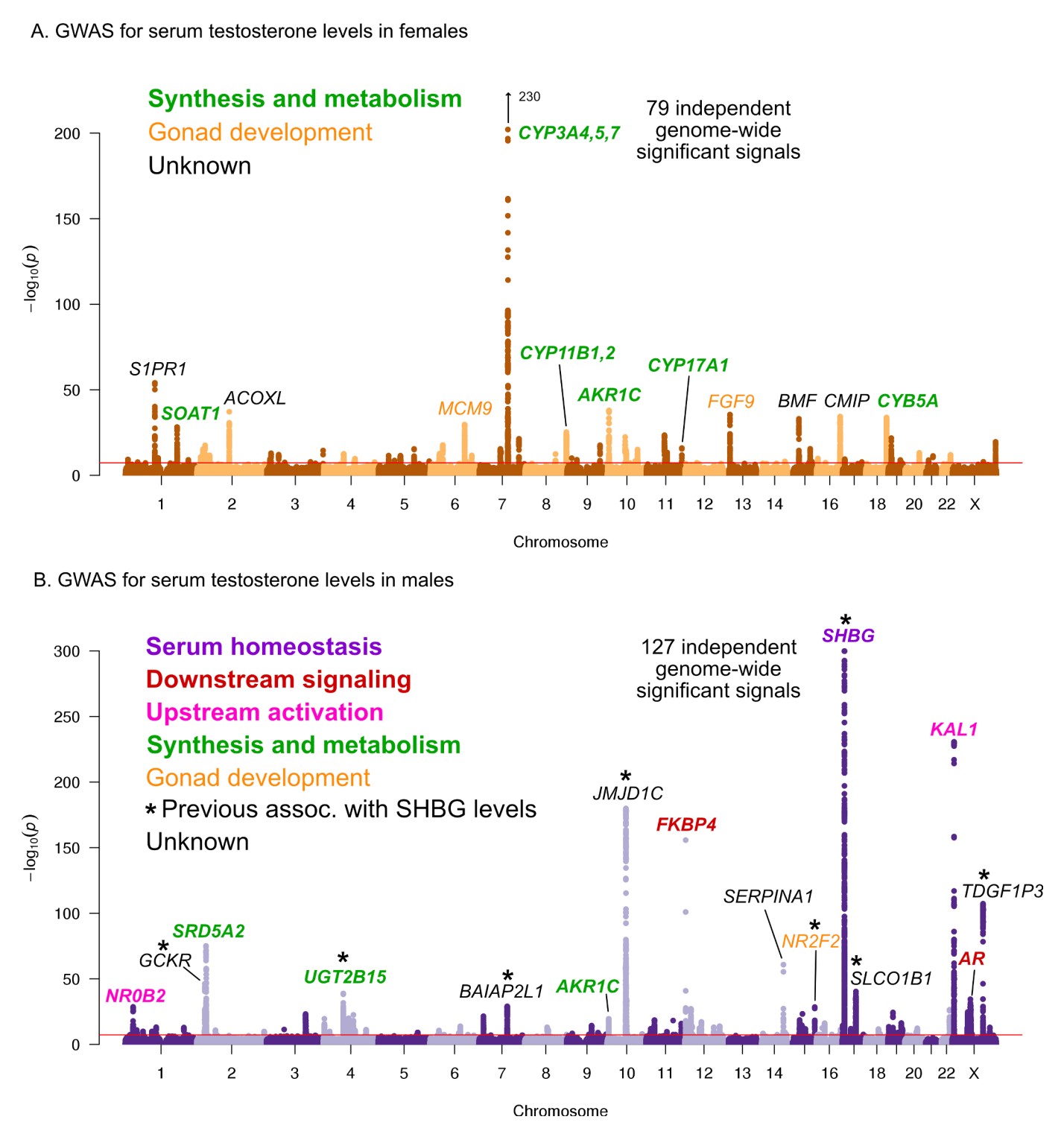

**Figure 5.** Manhattan plots for testosterone. (**A**) Females. (**B**) Males. Notice the low overlap of lead signals between females and males. FAM9A and FAM9B have been previously proposed as the genes underlying the KAL1 locus (*Ohlsson et al., 2011*).

The online version of this article includes the following figure supplement(s) for figure 5:

**Figure supplement 1.** Distributions of female and male luteinizing hormone, testosterone, sex hormone binding globulin (SHBG), and calculated bioavailable testosterone (CBAT) levels in the UK Biobank.

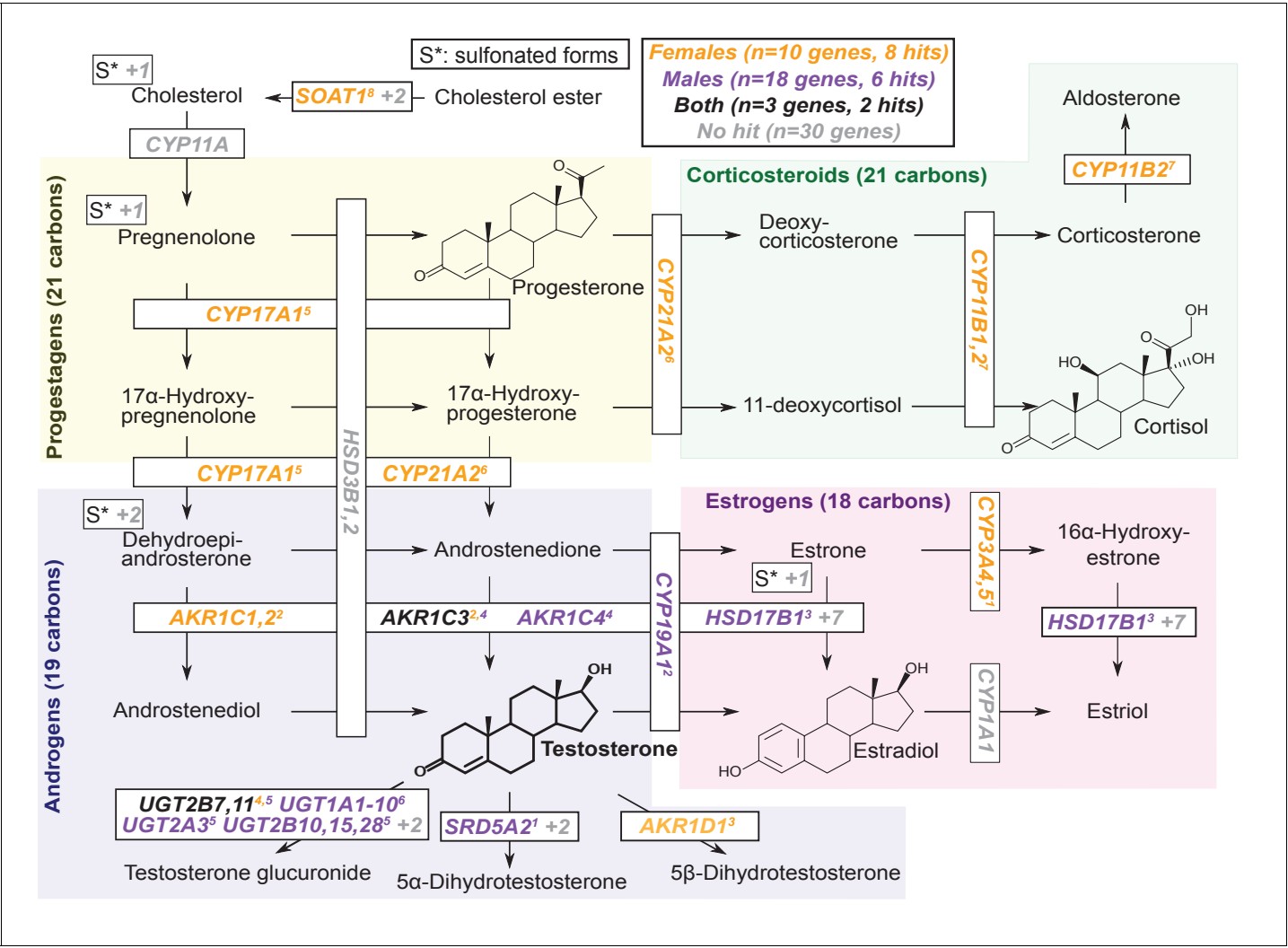

**Figure 6.** Pathway diagram for steroid hormone biosynthesis showing GWAS hits for females and males. The text color indicates genes within 100 kb of a genome-wide significant hit for females (orange), males (blue), or both females and males (black). Gray gene names or numbers indicates genes with no hits. Colored superscripts indicate multiple genes from the same locus (and hence may reflect a single signal). 'S*' indicates that an additional, sulfonated metabolite, along with the catalytic step and enzymes leading to it, is not shown. Pathway from KEGG; simplified based on a similar diagram in *Wikipedia, 2012*.

The online version of this article includes the following figure supplement(s) for figure 6:

**Figure supplement 1.** The KEGG pathway for steroid hormone biosynthesis is enriched for hits in both female and male testosterone GWAS.

**Figure supplement 2.** Non-overlapping female and male GWAS signals at AKR1C (left) and PDE2A (right) loci.

out of 61 genes are within 100 kb of a genome-wide significant signal in males, females or both. Indeed, the KEGG steroid hormone pathway shows strong enrichment for signals in both females and males (26-fold enrichment, p=2.5e-8 in females; 11-fold enrichment, p=1.2e-4 in males; *Figure 6—figure supplement 1*). While this pathway shows clear enrichment in both females and males, the major hits do not overlap. At two loci, AKR1C and PDE2A, female and male hits co-occur at the same locus, but are localized to different SNPs (*Figure 7—figure supplement 1*). More broadly, male hits and female hits tend to occur in different parts of the steroid hormone biosynthesis pathway: catalytic steps involved in progestagen and corticosteroid synthesis and metabolism only showed hits in females, while most male hits were concentrated within androgen synthesis, either upstream or downstream of testosterone itself (*Figure 6*).

## Genetics of testosterone regulation in males versus females

One remarkable feature of the testosterone data is the lack of sharing of signals between females and males. This is true for genome-wide significant hits, for which there is no correlation in the effect sizes among lead SNPs (*Figure 7A*), as well as genome-wide, as the global genetic correlation between females and males is approximately zero (*Figure 7—figure supplement 1*).

As we show below, two aspects of testosterone biology can explain these extreme sex differences in genetic architecture. First, the hypothalmic-pituitary-gonadal (HPG) axis plays a more significant role in regulating testosterone production in males than in females. This is due to sex differences in both endocrine signaling within the HPG axis and the tissue sources of testosterone production. Second, SHBG plays an important role in mediating the negative feedback portion of the HPG axis in males but not in females.

To assess the role of HPG signaling, we searched for testosterone GWAS hits involved in the transmission of feedback signals through the hypothalamus and pituitary (*Figure 7B*, genes reviewed in *Skorupskaite et al., 2014*). We also considered hits from GWAS of calculated bioavailable testosterone (CBAT), which refers to the non-SHBG-bound fraction of total teststerone that is free or albumin-bound, and can be inferred given levels of SHBG, testosterone, and albumin and assuming experimentally determined rate constants for binding (*Vermeulen et al., 1999*). CBAT GWAS thus controls for genetic effects on total testosterone that are mediated by SHBG production.

We found hits for both male testosterone and male CBAT throughout the HPG signaling cascade (*Figure 7B*). These include genes involved in the direct response of the hypothalamus to testosterone (AR, FKBP4) (*Smith et al., 2005*); modulation of the signal by either autoregulation (TAC3, TACR3) (*Skorupskaite et al., 2014*) or additional extrinsic endocrine signals (LEPR) (*Ahima et al., 1996*; *Barash et al., 1996*); downstream propagation (KISS1) (*Messager et al., 2005*) and the development of GnRH-releasing neurons in the hypothalamus (KAL1, CHD7) (*Cariboni et al., 2004*; *Layman et al., 2011*); and LH-releasing gonadotropes in the pituitary (GREB1) (*Li et al., 2017*). All these hits showed more significant effects on CBAT as compared to total testosterone (*Figure 7—figure supplement 3*), suggesting that their primary role is in regulating bioavailable testosterone.

Importantly, these HPG signaling hits do not show signals in females. To further investigate the different roles of the HPG axis in males versus females, we performed GWAS of LH levels using UKBB primary care data (N = 10,255 individuals). (Recall that LH produced by the pituitary signals to the gonads to promote sex hormone production.) If HPG signaling is important for testosterone production in males but not females, variants affecting LH levels should affect testosterone levels in males but not females. Consistent with this, we found significant positive genetic correlation between LH and male but not female testosterone (male $r_g = 0.27$, $p = 0.026$; female $r_g = 0.084$, $p = 0.49$; *Figure 7C*). These results were similar when considering measured testosterone and LH levels rather than genetic components thereof (*Supplementary file 5*).

Two known features of the HPG axis can explain the lack of association in females. First, the adrenal gland, which is not subject to control by HPG signaling, produces ~50% of serum testosterone in females. Consistent with this idea, GWAS hits for female testosterone cluster in steroid hormone pathways involving progestagen and corticosteroid synthesis (*Figure 6*), processes known to occur largely in the adrenal. Female testosterone hits are also specifically enriched for high expression in the adrenal gland relative to male testosterone hits (*Figure 7—figure supplement 4*).

Second, for the ovaries, which produce the remaining ~50% of serum testosterone in females, the net effect of increased LH secretion on testosterone production is expected to be diminished. This is because the pituitary also secretes follicle-stimulating hormone (FSH), which in females stimulates aromatization of androgens (including testosterone) into estrogens (*Ulloa-Aguirre and Michael Conn, 2014*). In males, FSH does not stimulate androgen aromatization but is instead required for sperm production. Consistent with differential roles of FSH, a previously described GWAS hit for menstrual cycle length at FSHB (*Laisk et al., 2018*) shows suggestive association with testosterone in females but not males (*Supplementary file 6*).

In addition to the role of HPG signaling, the presence of many SHBG-associated variants among the top hits in male testosterone suggests that SHBG also underlies many of the sex-specific genetic effects (*Figure 5B*). We found high positive genetic correlation between female and male SHBG, as well as between SHBG and total testosterone in males but not females (*Figure 7C*). Additionally, we found a significant negative genetic correlation between SHBG and CBAT in both females and

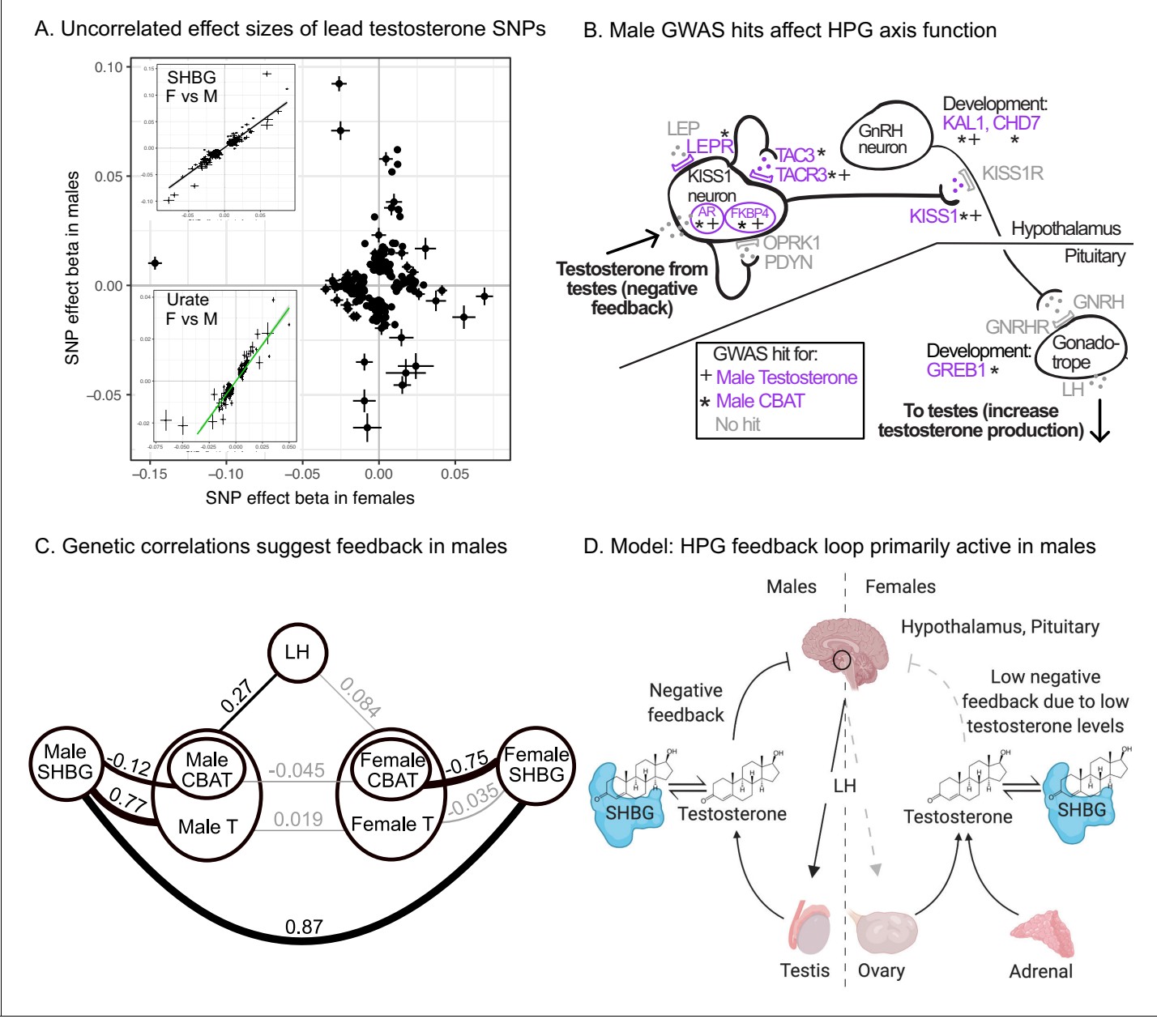

**Figure 7.** Sex differences in genetic variation in testosterone. (A) When comparing lead SNPs (p<5e-8 ascertained in either females or males), the effects are nearly non-overlapping between females and males. Other traits show high correlations for the same analysis (see urate and SHBG in inset). (B) Schematic of HPG axis signaling within the hypothalamus and pituitary, with male GWAS hits highlighted. These variants are not significant in females. (C) Global genetic correlations, between indicated traits (estimated by LD Score regression). Thickness of line indicates strength of correlation, and significant (p<0.05) correlations are in bold. Note that LH genetic correlations are not sex-stratified due to small sample size in the UKBB primary care data (N = 10,255 individuals). (D) Proposed model in which the HPG axis and SHBG-mediated regulation of testosterone feedback loop is primarily active in males. Abbreviations for all panels: SHBG, sex hormone-binding globulin; CBAT, calculated bioavailable testosterone; LH, luteinizing hormone.

The online version of this article includes the following figure supplement(s) for figure 7:

**Figure supplement 1.** Genetic correlations between females and males across select traits.

**Figure supplement 2.** Genetic correlations (estimated by LD Score Regression) between total testosterone, SHBG, and calculated bioavailable testosterone (CBAT) in females and males.

**Figure supplement 3.** Manhattan plot of difference in significance of assocation comparing GWAS of calculated bioavailable testosterone (CBAT) to total testosterone.

*Figure 7 continued on next page*

*Figure 7 continued*

**Figure supplement 4.** Mean expression of testosterone GWAS hits in females or males (defined as mean log-transformed counts of hits divided by mean log-transformed counts of all genes) in each of 48 GTEx tissues.

**Figure supplement 5.** Enrichment of random matched SNPs in core pathways.

males, but of a far larger magnitude in females than males (*Figure 7C*). Together, these observations suggest that while SHBG regulates the bioavailable fraction of testosterone in the expected manner in both females and males, there is subsequent feedback in males only, where decreased CBAT leads to increased total testosterone.

Combining these observations, we propose that increased SHBG leads to decreased bioavailable testosterone in both females and males, and in males this relieves the negative feedback from testosterone on the hypothalamus and pituitary gland, ultimately allowing LH production and increased testosterone production (*Figure 7D*). The lack of SHBG-mediated negative feedback in females is likely due in part to the overall weaker action of the HPG axis, as well as the fact that female testosterone levels are too low to effectively inhibit the HPG axis. This idea is supported experimental manipulations of female testosterone, which result in significant reductions of LH only when increasing testosterone levels to within the range typically found in males (*Serafini et al., 1986*).

In summary, we find that many of the top signals for female testosterone are in the steroid biosynthesis pathway, and a smaller number relate to gonadal development. In contrast, the lead hits for male testosterone reflect a larger number of processes, including especially SHBG levels and signaling components of the HPG axis, in addition to biosynthesis and gonadal development. These differences in the genetic architecture of female and male testosterone are so extreme that these can be considered unrelated traits from a genetic point of view.

## Polygenic architecture of the three traits

We have shown that the lead signals for all three traits are highly concentrated near core genes and core pathways. As an additional confirmation of these enrichments, we found that (i) random sets of SNPs matched to urate, IGF-1, or testosterone GWAS hits showed far lower overlap with the corresponding core pathways (*Figure 7—figure supplement 5*), and (ii) an alternative approach (*de Leeuw et al., 2015*) showed highly similar gene-set enrichments to those we observed (*Supplementary file 7*). Given this observation, we wondered whether these traits might be genetically simpler than typical complex diseases–most of which are highly polygenic, and for which the lead pathways contribute relatively little heritability (*Boyle et al., 2017*; *Shi et al., 2016*).

To address this, we first estimated how much of the SNP-based heritability is explained by variation at genes in enriched pathways (see *Supplementary files 8–10* for pathways and genes used). We used HESS to estimate the SNP-based heritability in each of 1701 approximately-independent LD blocks spanning the genome (*Shi et al., 2016*; *Berisa and Pickrell, 2016*). Plotting the cumulative distribution of SNP-based heritability across the genome revealed that, across all four traits, most of the genetic variance is distributed nearly uniformly across the genome (*Figure 8A*).

In aggregate, core genes contribute modest fractions of SNP-based heritability, with the exception of the SLC2A9 locus, which HESS estimates is responsible for 20% of the SNP-based heritability for urate. Aside from this outlier gene, the core pathways contribute between approximately 1–11 percent of the SNP-based heritability.

## Numbers of causal variants

We next sought to estimate how many causal variants are likely to contribute to each trait (*Zhang et al., 2018*; *Frei et al., 2019*; *O'Connor et al., 2019*). This is fundamentally a challenging problem, as most causal loci have effect sizes too small to be confidently detected. As a starting point we used ashR, which is an empirical Bayes method that estimates the fraction of non-null test statistics in large-scale experiments (*Stephens, 2017*). As described previously, we stratified SNPs from across the genome into bins of similar LD Score; we then used ashR to estimate the fraction of non-null associations within each bin (*Boyle et al., 2017*). (For this analysis, we used the 2.8M SNPs with MAF >5%.) We interpret this procedure as estimating the fraction of all SNPs in a bin that are in LD with a causal variant.

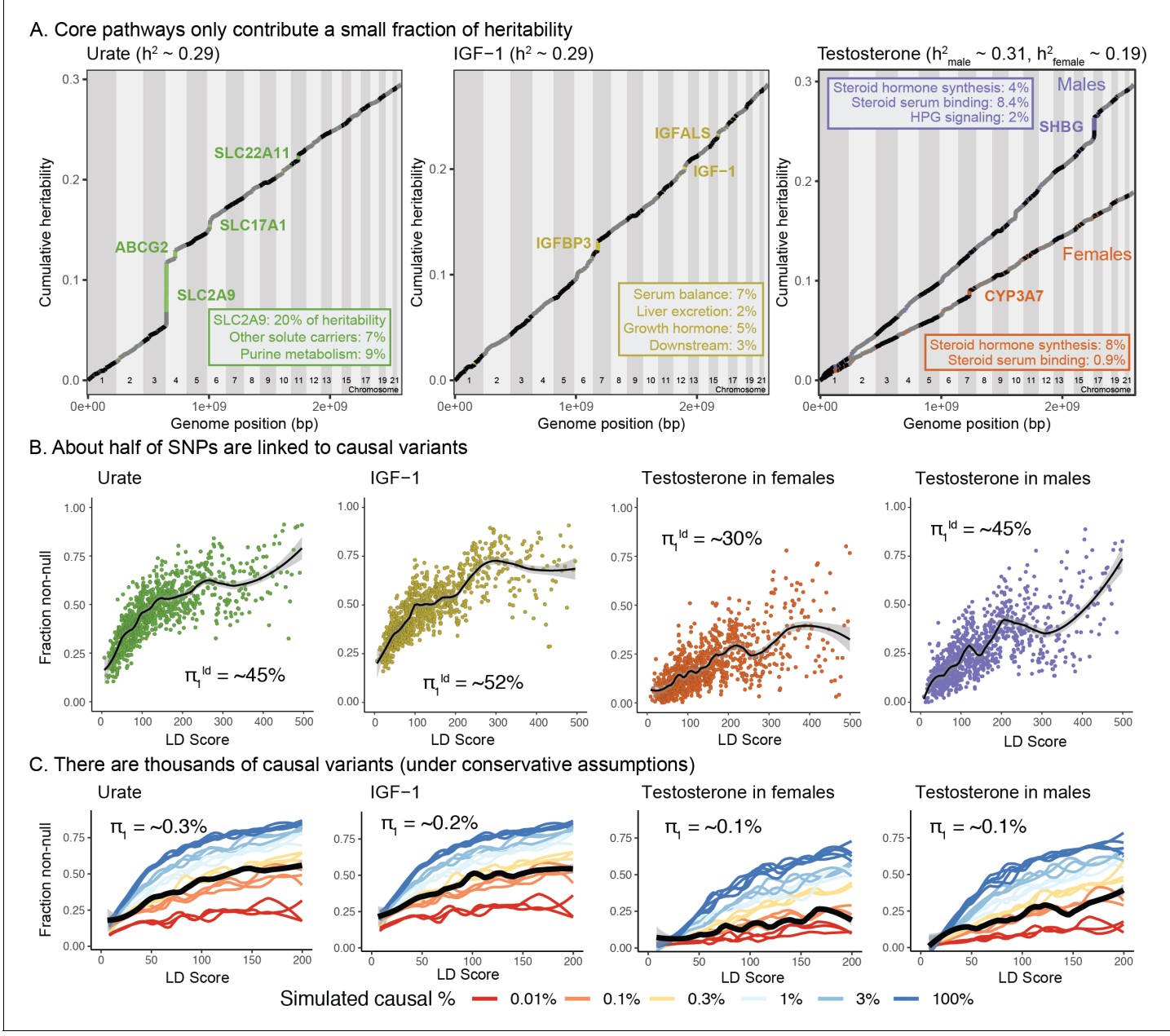

**Figure 8.** Despite clear enrichment of core genes and pathways, most SNP-based heritability for these traits is due to the polygenic background. (**A**) Cumulative distribution of SNP-based heritability for each trait across the genome (estimated by HESS). The locations of the most significant genes are indicated. Insets show the fractions of SNP-based heritability explained by the most important genes or pathways for each trait. (**B**) Estimated fractions of SNPs with non-null associations, in bins of LD Score (estimated by ashR). Each point shows the ashR estimate in a bin representing 0.1% of all SNPs. The inset text indicates the estimated fraction of variants with a non-null marginal effect, that is, the fraction of variants that are in LD with a causal variant. (**C**) Simulated fits to the data from (**B**). X-axis truncated for visualization as higher LD Score bins are noisier. Simulations assume that $\pi_1$ of SNPs have causal effects drawn from a normal distribution centered at zero (see Materials and methods). The simulations include a degree of spurious inflation of the test statistic based on the LD Score intercept. Other plausible assumptions, including clumpiness of causal variants, or a fatter-tailed effect distribution would increase the estimated fractions of causal sites above the numbers shown here.

The online version of this article includes the following figure supplement(s) for figure 8:

**Figure supplement 1.** Proportion of non-null associations in a random sample of 100,000 variants for each trait.

**Figure supplement 2.** Additional traits to fit causal simulations on.

**Figure supplement 3.** Prediction plots for the causal SNP counts underlying calculated bioavailable testosterone (CBAT) in females and males, as well as sex hormone binding globulin (SHBG) and a randomized version of SHBG.

**Figure supplement 4.** Parametric causal fraction on LD Scores reproduces SNP-based heritability-based estimates.

*Figure 8 continued on next page*

*Figure 8 continued*

**Figure supplement 5.** Estimates of causal sites are conservative with respect to SNP concentration within the genome.
**Figure supplement 6.** Effect of distribution of causal site betas on estimates of causal variant count.
**Figure supplement 7.** Association between minor allele frequency and estimated proportion of causal variants.
**Figure supplement 8.** Effect of minor allele frequency cutoff on the estimates obtained.
**Figure supplement 9.** Parametric causal fraction is robust to population structure.
**Figure supplement 10.** Estimating the effect of inflation mis-specification on the estimated causal variant count.
**Figure supplement 11.** Effect of mis-specification of SNP-based heritability or sample size in the simulation matching approach.
**Figure supplement 12.** Effect of GWAS covariates on estimates.
**Figure supplement 13.** Effect of bin count on estimates of causal variants.
**Figure supplement 14.** Distributions of (left) total SNP-based heritability of gene expression, or (right) fraction of expression SNP-based heritability driven by cis-effects (*Ouwens et al., 2020*) for genes in the indicated core pathways, or for all other MsigDB genes not in a core pathway.

For each trait, the fraction of non-null tests increases from low levels in the lowest LD Score bins to above 50% in the highest LD Score bins. Overall we estimate that around 45–50% of SNPs are linked to a non-zero effect variant for urate, IGF-1 and male testosterone, and 30% for female testosterone (*Figure 8B*). These estimates were robust to halving the sample size of the input GWAS, and were substantially higher than for randomized traits (simulated by permuting the IGF-1 and urate phenotypes) (*Figure 8—figure supplement 1*).

We next conducted simulations to understand how these observations relate to the numbers of causal variants (*Figure 8C*). To make this identifiable, we assume that a fraction $1 - \pi_1$ of all SNPs have an effect size that is exactly zero, while the remaner ($\pi_1$) draw their effect size from a single normal distribution with mean zero. Our goal is to estimate $\pi_1$. We simulated phenotypes for the UK Biobank individuals assuming a range of values of $\pi_1$ (Materials and methods). Causal variants were chosen uniformly at random from among the 4.4M SNPs with MAF >1%; effect sizes were simulated from a normal distribution with mean zero, and variances set to produce the observed SNP heritabilities (0.3 for urate, IGF-1, and male testosterone, and 0.2 for female testosterone). We also allowed for a degree of over-inflation of the test statistics (i.e. allowing for an inflation factor as in Genomic Control [*Devlin and Roeder, 1999*])–this was important for fitting the positive ashR estimates at low LD Scores. We then matched the simulations to the observed ashR results to approximate the numbers of causal variants.

Overall, our estimates range from 0.1% of all 4.4M variants with MAF >1% in female and male testosterone (~4000 causal sites) to 0.3% of variants for urate (~12,000 causal sites). These results imply that all four traits are highly polygenic, though considerably less so than height (for which we estimate 2%, or 80,000 causal sites in UK Biobank; *Figure 8—figure supplements 2* and *4*).

Furthermore, there are three reasons to suspect that these numbers may be underestimates. First, causal variants are likely to be clumped in the genome instead of being uniformly distributed; simulations with clumping require a larger number of causal variants to match the data (*Figure 8—figure supplement 5*). Second, if the distribution of effect sizes has more weight near zero and fatter tails than a normal distribution, this would imply a larger number of causal variants (see analysis assuming a T-distribution, *Figure 8—figure supplement 6*). Third, stratified LD Score analysis of the data suggests that some of the apparent evidence for overinflation of the test statistics (*Supplementary file 11*) may in fact be due to a higher proportion of causal variants occurring in lower LD Score bins (*Gazal et al., 2017*) rather than population stratification, as the annotation-adjusted intercepts for all traits but height are consistent with 1 (no population stratification).

We note that the proportion of causal variants estimated by ashR is substantially lower in low-MAF bins, even in infinitesimal models, presumably due to lower power (*Figure 8—figure supplements 7* and *8*). We overcame this by using a parametric fit, which is robust to inflation of test statistics (*Figure 8—figure supplements 9* and *10*); the resulting estimates were relatively similar, albeit slightly higher, than when using the simulation-matching method (*Figure 8—figure supplement 4*). We note that it is still critical to match samples by heritability and sample size, as in the simulation method (*Figure 8—figure supplement 11*), and to use correct covariates in the GWAS (*Figure 8—figure supplement 12*).

As an alternative approach, we used the program GENESIS, which uses a likelihood model to fit a mixture of effect sizes using 1–2 normal components, and a null component (*Zhang et al., 2018*;

*Supplementary file 12*). Assuming a single normal distribution, the results for the molecular traits were very similar to our results: male testosterone 0.1%; female testosterone 0.2%; urate 0.3%; IGF-1 0.4%. The GENESIS results for a mixture of two normal distributions resulted in a significantly higher overall likelihood, and estimates roughly threefold higher than our estimates: male testosterone 0.6%; female testosterone 0.7%; urate 1.1%; IGF-1 1.1%. GENESIS estimates for height were lower than ours (0.6% and 1.2%, respectively); it is possible that there is a downward bias at high polygenicity as GENESIS estimates for a simulated fully infinitesimal model were 2.7%.

In summary this analysis indicates that for these molecular traits, around 10–15% of the SNP-based heritability is due to variants in core pathways (and in the case of urate, SLC2A9 is a major outlier, contributing 20% on its own). However, most of the SNP-based heritability is due to a much larger number of variants spread widely across the genome, conservatively estimated at 4000–12,000 common variants for the biomarkers and 80,000 for height.

## Discussion

In this study, we examined the genetic basis of three molecular traits measured in blood serum: a metabolic byproduct (urate), a signaling protein (IGF-1), and a steroid hormone (testosterone). We showed that unlike most disease traits, these three biomolecules have strong enrichments of genome-wide significant signals in core genes and related pathways. At the same time, other aspects of the data are reminiscent of patterns for complex common diseases, including high polygenicity, little indication of allelic dominance or epistasis, and enrichment of signals in tissue-specific regulatory elements spread across the genome.

Our main results are as follows.

- Urate: The largest hits for urate are in solute carrier genes in the kidneys that shuttle urate in and out of the blood and urine. Remarkably, eight out of ten annotated urate transporters have genomewide significant signals. A single locus, containing SLC2A9, is responsible for 20% of the SNP-based heritability. While the urate transport pathway was previously known to be enriched in GWAS hits (*Tin et al., 2019*), we further demonstrate that the purine biosynthetic pathway, from which urate is produced as a byproduct, is modestly enriched for signals (2.1-fold). Several master regulators for kidney and liver development are among the most significant hits. Aside from SLC2A9, the overall SNP-based heritability is primarily driven by variants in kidney regulatory regions, both shared across cell types and not.

- IGF-1: IGF-1 is a key component of a signaling cascade that links growth hormone released from the pituitary to stimulation of cell growth in peripheral tissues. We identified 354 independent genome-wide significant signals. The strongest signals lie in genes that interact directly with IGF-1, including IGFBP3, as well as in the IGF1 gene itself. More generally, we see striking enrichment of hits throughout the growth hormone-IGF cascade–this includes especially the upper parts of the cascade, which regulate IGF-1 release, but also in downstream components of the cascade as well, suggesting a feedback mechanism on IGF-1 levels. These pathway-level enrichments were not identified in previous, less well-powered GWAS of IGF-1 levels (*Teumer et al., 2016*).

- Testosterone: In contrast to urate, testosterone shows clear enrichment of signals within the steroid biosynthesis pathway (26-fold in females, 11-fold in males). Remarkably, the genetic basis of testosterone is almost completely independent between females and males, as reported recently (*Flynn et al., 2021*; *Ruth et al., 2020*). In females, the lead hits are mostly involved in synthesis. In males, in addition to hits in the synthesis pathway, we see signals throughout the hypothalamic-pituitary-gonadal (HPG) axis which regulates testosterone production in the gonads, as well as in variants that regulate SHBG. Furthermore, in males, increased SHBG reduces negative feedback between testosterone levels and the HPG axis, thereby increasing total serum testosterone. These results provide a mechanistic explanation of the sex differences in testosterone genetics, in addition to showing that GWAS hits can reveal the core biology of a trait even in the context of vastly differing genetic architecture between the sexes.

- Polygenic background. For each of these traits, the core genes and pathways contribute only a modest fraction of the total SNP-based heritability. Aside from SLC2A9 for urate, the most important core pathways contribute up to about 10% of the total SNP-based heritability. We estimated the numbers of causal variants under a model where causal variants have a normal effect-size distribution. We estimate that there are around 4000–12,000 common variants with

non-zero effects on these traits. Using the same method, we estimated about 80,000 causal sites for height. These estimates are likely conservative as several of our assumptions may lead us to underestimate the true values.

## Understanding the architecture of complex traits

One of our motivations in this study was to use these three traits as models to extend our understanding of the genetic architecture and types of genes underlying complex traits.

Many of the advances of 20th century genetics resulted from focused study of the functions of major-effect mutations; this principle has been extended in the GWAS era into interpreting the impact of lead signals. And yet, at the same time, most heritability is driven by the polygenic background of small effects at genes that are not directly involved in the trait. The overwhelming importance of the polygenic background of thousands of small effects is a striking discovery of modern GWAS, and demands explanation as it does not fit neatly into standard conceptual models of the relationship between genotype and phenotype.

As discussed in the Introduction, our group recently proposed a simplified conceptual model to understand this phenomenon (*Boyle et al., 2017*; *Liu et al., 2019*). We proposed that for any given trait there is a limited set of core genes that are directly involved in the biology of the phenotype, but that most of the heritability is due to SNPs with cis-effects on other (peripheral) genes that are expressed in the same tissues; these in turn affect core genes via trans-regulatory networks. Thus far, it has been difficult to fully test this model because, in general, we do not know many of the genes that may have direct effects in any given trait. We also generally have very limited knowledge of trans-regulatory networks.

The present paper helps to fill part of this gap by studying the genetic basis of three molecular traits where we can a priori identify a large number of core genes and their associated pathways. Thus, our work provides concrete examples of how genetics can affect core biology to an extent that is usually difficult to achieve for disease traits. Furthermore, consistent with the model and our previous analyses of gene expression (*Liu et al., 2019*), we find that the known core pathways contribute only a modest fraction of the SNP-based heritability, and that the bulk of the heritability is driven by thousands of variants spread across much of the genome.

That said, it remains difficult to test the second part of the model, that is, that most of the heritability passes through trans-regulatory networks. This problem is challenging because we currently lack the sample sizes necessary for inferring trans-regulatory networks in any tissue or cell type, with the possible exception of whole blood. Secondly, its likely that the relevant networks may be active only in particular cell types or at particular times in development, which makes the estimation of trans-regulatory networks even more difficult. However, one promising approach has recently yielded results consistent with the trans-regulatory part of the omnigenic model. Võsa et al have shown that genome-wide polygenic scores for several traits correlate with the expression levels of core genes for those traits, as would be predicted by the model (*Võsa et al., 2018*). Nonetheless, the trans-regulatory component of the model requires further work.

Another type of explanation for high polygenicity comes from the observation that many traits and diseases are affected by multiple biological processes. Thus, any variants that affect those intermediate processes can potentially be detected in GWAS of the endpoint trait (*Turkheimer, 2000*; *Gottesman and Gould, 2003*; *Bittante et al., 2012*; *Pickrell et al., 2016*; *Udler, 2019*). While this process undoubtedly contributes to the polygenicity of many endpoint traits, our data suggest it is unlikely that this type of process drives high polygenicity for these molecular traits. Notably, for urate, we estimated ~12,000 causal variants, and showed that the vast majority of the SNP-based heritability likely acts through the kidneys. Thus, any explanation for the high polygenicity of urate must presumably depend on the role of genetic variation on kidney function in general, and urate transport in particular.

The huge polygenicity of complex traits also raises questions about how to extract biological insight from GWAS. If there are tens of thousands of associated variants, acting through thousands of genes, then presumably most of these will not be especially helpful for understanding mechanisms of disease (*Goldstein, 2009*). (In contrast, for constructing polygenic scores, we do in fact care about all variants, as small effects drive most of the phenotypic variance.) This raises the question of how to use GWAS to identify the genes that are actually most proximal to function. This is of course

a question that many in the field have wrestled with, for a wide variety of traits (*de Leeuw et al., 2015*; *Pers et al., 2015*). Overall, we can expect that the most significant variants will usually point to biologically important genes for the corresponding trait. That said, there are many reasons why significance is not a fully reliable indicator of gene importance: significance depends on both the variant effect size and its allele frequency; the allele frequency is a random outcome of genetic drift and, moreover, selection tends to lower frequencies of the most important variants (*Simons et al., 2018*; *O'Connor et al., 2019*); lastly the effect size of the variant depends not only on the importance of the gene for the trait, but also on the magnitude of that variant's effect on the gene (e.g. as a cis-eQTL). Furthermore, some genes that are biologically important may be entirely missed because they do not happen to have common functional variants. Nonetheless, given all these caveats, we found that for these three molecular traits the lead GWAS hits were indeed highly enriched for core genes, consistent with work for other traits where many of the lead variants are interpretable (*Lu et al., 2017*; *Liu et al., 2017*; *de Lange et al., 2017*).

In summary, we have shown that for three molecular traits, the lead hits illuminate core genes and pathways to a degree that is highly unusual in disease or complex trait GWAS. By doing so they illustrate which processes may be most important for trait variation. For example, for urate, kidney transport is more important than biosynthesis, while for testosterone, biosynthesis is important in both sexes but especially in females. However, in other respects, the GWAS data here are reminiscent of more-complex traits: in particular most trait variance comes from a huge number of small effects at peripheral loci. These vignettes help to illustrate the architecture of complex traits, with lead variants that are directly involved in trait biology alongside a massively polygenic background.

## Materials and methods

### Population definition

We defined our GWAS population as a subset of the UK Biobank (*Bycroft et al., 2018*). We use ~337,000 unrelated White British individuals as our cohort, filtering based on sample QC characteristics as previously described (*Sinnott-Armstrong et al., 2021*):

1. Used to compute principal components (`used_in_pca_calculation` column).
2. Not marked as outliers for heterozygosity and missing rates (`het_missing_outliers` column).
3. Do not show putative sex chromosome aneuploidy (`putative_sex_chromosome_aneuploidy` column).
4. Have at most 10 putative third-degree relatives (`excess_relatives` column).
5. Finally, we used the `in_white_British_ancestry_subset` column in the sample QC file to define the subset of individuals in the White British cohort.

### Trait definition

We perform trait normalization and quality control similarly to previous work (*Sinnott-Armstrong et al., 2021*). Trait measurements are first log-transformed, then adjusted for genotype principal components, age indicator variables, sex, 5 year age ('approximate age') by sex interactions, self-identified ethnicity, self-identified ethnicity by sex interactions, fasting time, estimated sample dilution factor, assessment center, genotyping batch, 20-tile of time of sampling, month of assessment, and day of assay.

Then, individuals were subset to the GWAS population (defined above), separated by sex for testosterone measurements. The final sample sizes were 318,526 for urate, 317,114 for IGF-1, 142,778 for female testosterone, and 146,339 for male testosterone.

### GWAS

We performed GWAS in plink2 alpha using the following command (data loading arguments removed for brevity):

```
plink2 -glm cols=chrom,pos,ref,alt,alt1,ax,a1count,totallele,a1freq,
       machr2,firth,test,nobs,beta,se,ci,tz,p omit-ref
    -covar-variance-standardize
```

```
-remove [non-White-British, related White British or excluded]
-keep [male, female, or all]
-geno 0.2 -hwe 1e-50 midp -threads 16
```

GWAS were then filtered to observed allele frequency greater than 0.001 and INFO score greater than 0.3 for further analyses.

## GWAS for paired difference epistasis

A GWAS was performed in two subsets of individuals – those with two C alleles at rs16890979 (N = 295209) and those with two T alleles at rs16890979 (N = 30184). The following command was used:

```
plink2 -glm cols=chrom,pos,ref,alt,a1freq,firth,test,
        nobs,beta,se,ci,tz,p hide-covar
   -hwe 1e-50 midp -keep [rs16890979 CC or TT individuals]
   -remove [non-White British] -geno 0.1 -maf 0.001
```

With covariates including adjusting for age, age squared, genotyping array, and 20 principal components. The residual urate levels, already adjusted for age, sex, global principal components, and technical covariates (Methods) were used as input.

After GWAS completed, SNPs valid in both CC and TT individuals were compared for betas using a paired difference Z test. The test statistic was then converted to a p-value using a standard normal distribution.

## LH trait definition

LH levels were extracted from UK Biobank primary care data using code XM0lv. Separately, LH levels extracted using code XE25I were also included for phenotypic correlation analyses. The median level across observations and log number of observations were recorded for covariate correction below. Individuals with median observations more than 10 times the interquartile range away from the median of medians were discarded. Once these individuals were removed, individuals with observations more than four standard deviations from the resulting mean were also discarded.

For the primary LH code XM0lv, the distribution of raw, cleaned, and covariate-adjusted phenotype values were respectively:

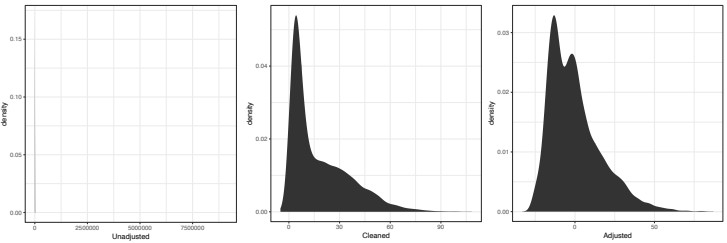

**Scheme 1.** Distribution of raw (left), cleaned (middle), and covariate-adjusted (right) phenotype values for primary luteinizing hormone (LH) code XMOlv.

For the secondary LH code XE25I, the distribution of raw, cleaned, and covariate-adjusted phenotype values were respectively:

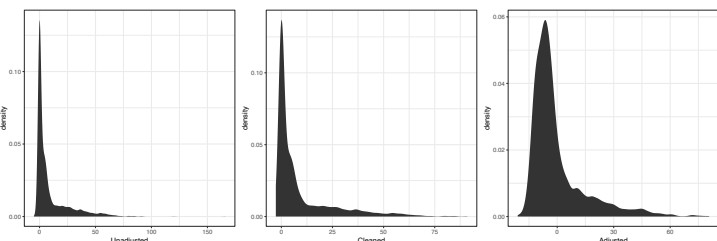

**Scheme 2.** Distribution of raw (left), cleaned (middle), and covariate-adjusted (right) phenotype values for secondary LH code XE25I.

For GWAS, the cleaned phenotypes were log-transformed and adjustments were used as covariates.

## LH GWAS

Age, sex, genotyping array, 10 PCs, log number of observations in primary care, and which primary care code produced a given observation were used as covariates.

We performed GWAS in plink2 alpha using the following command (data loading arguments removed for brevity):

```
plink2 –glm cols=chrom,pos,ref,alt,alt1,ax,a1count,totallele,a1freq,
     machr2,firth,test,nobs,beta,se,ci,tz,p hide-covar omit-ref
  –covar-variance-standardize
  –remove [non-White-British, related White British or excluded]
  –keep [all White British]
  –geno 0.2 –hwe 1e-50 midp –maf 0.005 –vif 999
```

We also performed GWAS of LH code XE25I in a sex stratified fashion using the following command:

```
plink2 –glm cols=chrom,pos,ref,alt,alt1,ax,a1count,totallele,
     a1freq,machr2,firth,test,nobs,beta,se,ci,tz,p
  hide-covar omit-ref –covar-variance-standardize –remove <non-White-British>
  –geno 0.2 –hwe 1e-50 midp –threads {threads} –maf 0.001 –vif 999;
```

On genotyped SNPs and imputed variants with a minor allele frequency greater than 1% in the White British as a whole.

GWAS were then filtered to MAF >1% and INFO >0.7. These higher threshold were chosen to reflect the much smaller sample size in the GWAS.

## GWAS hit processing

To evaluate GWAS hits, we took the list of SNPs in the GWAS and ran the following command using plink1.9:

```
plink –bfile [] –clump [GWAS input file] –clump-p1 1e-4 –clump-p2 1e-4
   –clump-r2 0.01 –clump-kb 10000 –clump-field P –clump-snp-field ID
```

We then took the resulting independent GWAS hits and examined them for overlap with genes. In addition, for defining the set of SNPs to use for enrichment analyses, we greedily merged SNPs located within 0.1 cM of each other and took the SNP with the minimum p-value across all merged lead SNPs. In this way, we avoided potential overlapping variants that were driven by the same, extremely large, gene effects.

## Gene proximity

We annotated all genes in any Biocarta, GO, KEGG, or Reactome MSigDB pathway as our full list of putative genes (in order to avoid pseudogenes and genes of unknown function), and included the genes within each corresponding pathway as our target set. This resulted in 17,847 genes. We extended genes by 100 kb (truncating at the chromosome ends) and used the corresponding regions, overlapped with SNP positions, to define SNPs within range of a given gene. Gene positions were defined based on Ensembl 87 gene annotations on the GRCh37 genome build.

## Pathway enrichment of GWAS hits

GWAS hit pathway enrichment was evaluated using Fisher's exact test. For each pathway for a given trait (*Supplementary files 8–10*), genes were divided into those within the pathway and those outside; and separately into genes within 100 kb of a GWAS hit and not. A $2 \times 2$ Fisher's exact test was used to estimate the total enrichment for GWAS hits around genes of interest.

For female and male testosterone, we noticed a number of GWAS loci with multiple paralogous enzymes within the synthesis pathway (e.g. *AKR1C*, *UGT2B*, *CYP3A*). To avoid double counting GWAS hits when testing enrichment at such loci, we instead considered the number of GWAS hits (within 100 kb of any pathway gene as above) normalized to the total genomic distance covered by all genes (±100 kb) in the pathway. A Poisson test was used to compare the rate parameter for this GWAS hit/Mb statistic between genes in a given pathway and all genes not in the pathway.

To quantify pathway enrichment expected from random sets of SNPs not associated with a phenotype, we used SNPSnap (*Pers et al., 2015*) with default settings to obtain 1000 sets of equally-sized random SNPs matched to urate, IGF-1, or testosterone hits in terms of LD, minor allele frequency, and genic distance. For each set of random, matched SNPs, we determined the number of core genes within 100 kb as for the true set of GWAS hits.

To quantify pathway enrichments using an alternative approach, we used MAGMA (*de Leeuw et al., 2015*) with a 10 kb gene window and with the default competitive mode. We tested enrichment for all gene sets in Biocarta, GO, KEGG, or Reactome MSigDB, as well as additional curated sets of core genes for the three traits.

## Partitioned heritability

Partitioned SNP-based heritability estimates were generated using LD Score regression (*Finucane et al., 2015*). The BaselineLD version 2.2 was used as a covariate, and the 10 tissue type LD Score annotations were used as previously described (*Finucane et al., 2015*) in a multiple regression setup with all cell type annotations and the baseline annotations.

## Pathway heritability estimation

We evaluated SNP-based heritability in pathways using two distinct strategies. Initially, we used partitioned LD Score regression (*Finucane et al., 2015*) but found that the estimates were somewhat noisy, likely because most pathways contain few genes. As such, we used alternative fixed-effect models for which there is increased power.

Next, we calculated the SNP-based heritability in a set of 1701 approximately independent genomic blocks spanning the genome (*Berisa and Pickrell, 2016*) using HESS (*Shi et al., 2016*). Next, we overlapped blocks with genes in each pathway. The SNP-based heritability estimates for all blocks containing at least one SNP within 100 kb of a pathway gene were summed to estimate the SNP-based heritability in a given pathway. Pathway definitions were assembled based on a combination of KEGG pathways, Gene Ontology categories, and manual curation based on relevant reviews.

## Causal SNP simulations

All imputed variants with MAF >1% in the White British (4.1M) were used as a starting set of putative causal SNPs. Individual causal variants were chosen at random, with a fraction $P$ of them marked as causal. Each causal variant was assigned an effect size:

$$\beta \sim \mathrm{N}(0, 1)$$

For our simulations, we used $P \in \{0.0001, 0.001, 0.003, 0.01, 0.03\}$.

Next, GCTA was used to simulate phenotypes based on the marked causal variants, using the following command:

```
gcta64 –simu-qt –simu-causal-loci CausalVariantEffects
    –simu-hsq 0.3 –bfile UKBBGenotypes"
```

Producing predicted phenotypes with SNP-based heritability $h^2 = 0.3$. GWAS were run within both the full set of 337,000 unrelated White British individuals and a randomly downsampled 50%, to approximate the sex-specific GWAS used for Testosterone, across the set of putative causal SNPs. GWAS for the traits, as well as a random permuting across individuals of urate and IGF-1 to act as negative controls, were repeated on this subset of variants as well. In this way, we have a directly comparable set of simulated traits to use, along with the corresponding true traits and negative controls, to ascertain causal sites in the genome.

For the infinitesimal simulations, instead plink was used to generate polygenic scores on the basis of the random assignment of effect sizes to SNPs, and these were then normalized with $N(0, \sigma^2)$ environmental noise such that $h^2$ was the given target SNP-based heritability.

### Causal SNP count fitting procedure using ashr

LD Scores for the 489 unrelated European-ancestry individuals in 1000 Genomes Phase III (*Bulik-Sullivan et al., 2015*) were merged with the GWAS results along with LD Scores derived from unrelated European ancestry participants with whole genome sequencing in TwinsUK. TwinsUK LD Scores are used for all analyses. Then variants were filtered by minor allele frequency to either greater than 1%, greater than 5%, or between 1% and 5%. Remaining variants were divided into 1000 equal sized bins, along with 5000 and 200 bin sensitivity tests. Within each bin, the ashR estimates of causal variants, as well as the mean $\chi^2$ statistics, were calculated using the following line of R:

```
data %>% filter(pmin(MAF, 1-MAF) > min.af, pmin(MAF, 1-MAF) < max.af) %>%
    mutate(ldBin = ntile(ldscore, bins)) %>% group_by(ldBin) %>%
    summarize(mean.ld = mean(ldscore), se.ld=sd(ldscore)/sqrt(n()),
        mean.chisq = mean(T_STAT**2, na.rm=T),
        se.chisq=sd(T_STAT**2, na.rm=T)/sqrt(sum(!is.na(T_STAT))),
        mean.maf=mean(MAF),
        prop.null = ash(BETA, SE)$fitted_g$pi[1], n=n())
```

Thus, the within-bin $\chi^2$ and proportion of null associations $\pi_0$ were each ascertained. Next, these fits were plotted as a function of `mean.ld` to estimate the slope with respect to LD Score, and true traits were compared to simulated traits, described below.

We use two fixed simulated heritabilities, $h^2 = 0.3$ and $h^2 = 0.2$, to approximately capture the set of heritabilites observed among our biomarker traits. Traits with true SNP-based heritability among variants with MAF >1% different than their closest simulation might have causal site count over-estimated (for $h^2_{true} > h^2_{sim}$) or under-estimated (for $h^2_{true} < h^2_{sim}$). In addition, most traits in reality have more than zero SNPs with MAF <1% contributing to the SNP-based heritability. Thus, we take these estimates as approximate and conservative.

### Effect of population structure on causal SNP estimation

We expect that population structure might lead to test statistic inflation for causal variant and genetic correlation estimates (*Berg et al., 2019*). To evaluate this, we performed GWAS for height using no principal components, and evaluated the causal variant count (*Figure 8—figure supplement 12*).

This suggests that the test statistic inflation is an important parameter in the estimation of causal variants, as is intuitive. As such, we generated estimated SNP counts for five different inflation values (0.9, 1, 1.05, 1.1, and 1.2) and plotted all of them, under the assumption that the best fitting intercept would have the most calibrated estimates. Plots are replicated across these intercepts in the sensitivity analyses shown, as in *Figure 8—figure supplement 9*.

## Evaluating the calibration of causal SNP proportion estimation

To evaluate calibration of causal SNP estimates, in addition to using simulated traits as the controls, we also generated a randomized control by shuffling the SHBG phenotype values across individuals (*Figure 8—figure supplement 3*). We performed this analysis using urate and IGF-1 to similar effect (data not shown).

This suggests that the causal variant counts are well calibrated for the randomized traits, even though they lack structure with respect to covariates.

## Effect of sample size on causal SNP estimation

It is important to note that these estimates are still likely power limited even in a study as large as UK Biobank. We make this note on the basis of observed $\pi_0$ for $MAF>5\%$ variants being uniformly higher than $1\%<MAF<5\%$ variants in both simulations and observed data for high causal variant counts (*Figure 8—figure supplement 8*).

As such, we anticipate that future studies with larger samples will yield increased, but asymptotic, estimates of causal SNP percentages among common variants, and treat our estimates as conservative bounds.

Particularly for height (*Figure 8—figure supplement 2*), while the uncalibrated estimates with the full sample are substantially higher than the half sample, the calibrated estimates are nearly identical. This suggests that trait polygenicity might be an important factor in determining the power of this method at different sample sizes, as height is known to be highly polygenic (*Shi et al., 2016*).

## Effect of binned variant count on causal SNP estimation

It is possible that the `ashR` algorithm itself, and not the GWAS, are the power limited step of the analysis. To evaluate this, we ran `ashR` on 200, 1000, and 5000 equally sized bins along the LD Score axis. We found that increasing bin counts both decrease the standard errors and the intercepts (*Figure 8—figure supplement 13*) and recommend as many bins as is practical.

## Effect of minor allele frequency on causal SNP estimation

Because we only simulated causal effects among SNPs with MAF >1%, we were concerned that variant effect bins might be biased by the minor allele frequency cutoff. We previously ran with higher MAF cutoffs (25% and 40%) as calibrations on an earlier version of the model, and observed uniformly larger causal SNP percentages. We saw relative robustness to lower thresholds, but overall the fraction of causal variants was lower in the lower MAF bins (*Figure 8—figure supplement 7*).

## Effect of concentrated SNPs on causal SNP estimation

For each variant, the megabase bin it is contained within was used as a proxy for SNPs in local LD. A within-megabase causal SNP percentage parameter:

$$P \sim \text{Beta}(\alpha, \alpha/\rho)$$

was chosen such that ρ was the overall expected percentage of causal sites in the genome across a concentration parameter α. For our simulations, we used $\rho \in \{0.0001, 0.0003, 0.001, 0.003, 0.01, 0.03, 0.05\}$ and $\alpha \in \{10, 3, 0.3\}$ to represent different degrees of 'clumpiness' along the genome.

## Genetic correlation between sex-stratified testosterone-related traits

LD Score regression [**Bulik-Sullivan2015-tx**] was used to generate genetic correlation estimates. The following command was used:

```
ldsc.py -rg <traits> -ref-ld-chr eur_ref_ld_chr
    -w-ld-chr eur_w_ld_chr
```

where `eur_*_ld_chr` were downloaded from https://data.broadinstitute.org/alkesgroup/LDSCORE/.

### Residual height comparison with IGF-1

Height (adjusted for age and sex) and residualized log IGF-1 levels for unrelated White British individuals were plotted against each other, and visualized using `geom_smooth`.

### Pathway diagrams

Diagrams were drawn using Adobe Illustrator and a Wacom graphics tablet.

### PheWAS analysis

PheWAS were performed using the Oxford Brain Imaging Genetics (BIG) Server (*Elliott et al., 2018*).

### Non-additivity tests

Residualized trait values were used as the outcome in all models. An ANOVA was performed between a model measuring the effect of genotype dosages versus a model with both genotype dosage effects and indicators for each rounded genotype. In this way, a large number of possible non-additive models are approximated with a single model. Analyses were performed in R 3.4 using `lm`.

### Epistasis tests

We estimated that for hits with p<1e-20 we would have power to detect interaction components that are at least 10% the magnitude of a main effect (see Materials and methods). Thus, we tested all pairwise interactions among the independent lead SNPs with p<1e-20. Residualized trait values were used as the outcome in all models. An ANOVA was performed between a model measuring the effect of indicators for each rounded genotype (4 degrees of freedom) versus the interaction between the two sets of indicators (8 degrees of freedom). In this way, a large number of possible non-additive models are approximated with a test. Alternative models with dominant-only effect interactions with fewer degrees of freedom were also tested with similar results. Analyses were performed in R 3.4 using `lm`.

### LD score regression for partitioning SNP-based heritability

We used partitioned LD Score regression (*Finucane et al., 2015*) to estimate the enrichment of individual tissues. We used the `ldsc` package and the updated BaselineLD v2.2 annotations with the following command:

```
ldsc.py –h2 <munged urate summary statistics> \
        –ref-ld-chr baselineLD.,<cell type annotations> \
        –overlap-annot –frqfile-chr 1000G_frq/1000G.mac5eur. \
        –w-ld-chr weights_hm3_no_hla/weights.
```

where `<cell type annotations>` were alternative either the default annotations for each of the ten cell type groups (*Finucane et al., 2015*) or modified versions which were filtered of any regulatory regions overlapping with the kidney cell type, using the following command:

```
ls 1000G_Phase3_cell_type_groups/*.bed | while read bed; do
   intersectBed –a $bed –b 1000G_Phase3_cell_type_groups/7.bed –v >
      1000G_Phase3_cell_type_groups_exclude_kidney/`basename $bed`;
done
```

In this way, the cell type exclusive, non-kidney regulatory elements are used.

## Acknowledgements

We thank members of the Pritchard, Page, Przeworski, Sella, and Bassik labs, as well as Ipsita Agarwal, Evan Boyle, Eric Fauman, Jake Freimer, Rebecca Harris, Yang Li, Xuanyao Liu, Iain Mathieson, Molly Przeworski, Guy Sella, Yuval Simons, and Jeff Spence for helpful discussions or comments; and

the UK Biobank and its participants for making this project possible, which we accessed through UK Biobank application number 24983. This work was supported by NIH grants HG008140 and HG009431 (to JKP), a Stanford Graduate Fellowship (to NS-A), a National Defense Science and Engineering Grant (to NS-A), and a Helen Hay Whitney Fellowship (to SN).

## Additional information

### Funding

| Funder | Grant reference number | Author |
|---|---|---|
| American Society for Engineering Education | NDSEG | Nasa Sinnott-Armstrong |
| Stanford University | Stanford Graduate Fellowship | Nasa Sinnott-Armstrong |
| National Human Genome Research Institute | HG008140 | Jonathan K Pritchard |
| National Human Genome Research Institute | HG009431 | Jonathan K Pritchard |
| Helen Hay Whitney Foundation | HHWF 2020 | Sahin Naqvi |

The funders had no role in study design, data collection and interpretation, or the decision to submit the work for publication.

### Author contributions

Nasa Sinnott-Armstrong, Conceptualization, Data curation, Software, Formal analysis, Supervision, Validation, Investigation, Visualization, Methodology, Writing - original draft, Project administration, Writing - review and editing; Sahin Naqvi, Data curation, Software, Formal analysis, Supervision, Validation, Investigation, Visualization, Methodology, Writing - original draft, Writing - review and editing; Manuel Rivas, Resources, Data curation, Writing - review and editing; Jonathan K Pritchard, Conceptualization, Resources, Supervision, Funding acquisition, Investigation, Visualization, Methodology, Writing - original draft, Project administration, Writing - review and editing

### Author ORCIDs

Nasa Sinnott-Armstrong (iD) https://orcid.org/0000-0003-4490-0601
Sahin Naqvi (iD) https://orcid.org/0000-0003-2635-7967
Jonathan K Pritchard (iD) https://orcid.org/0000-0002-8828-5236

### Ethics

Human subjects: This research has been conducted using the UK Biobank Resource under Application Number 24983, "Generating effective therapeutic hypotheses from genomic and hospital linkage data" (criteriahttp://www.ukbiobank.ac.uk/wp-content/uploads/2017/06/24983-Dr-Manuel-Rivas.pdf). Based on the information provided in Protocol 44532 the Stanford IRB has determined that the research does not involve human subjects as defined in 45 CFR 46.102(f) or 21 CFR 50.3(g). All participants of UK Biobank provided written informed consent (more information is available at https://www.ukbiobank.ac.uk/2018/02/gdpr/).

### Decision letter and Author response

Decision letter https://doi.org/10.7554/eLife.58615.sa1
Author response https://doi.org/10.7554/eLife.58615.sa2

## Additional files

### Supplementary files

• Supplementary file 1. Independent GWAS hits for urate. CHROM, chromosome number; POS, variant position (hg19); ID, variant identifier; REF, reference genome sequence allele; A1, alternative allele; A1_CT, number of A1 alleles; ALLELE_CT, total alleles; A1_FREQ, frequency of A1 allele; MACH_R2, estimated imputation accuracy (INFO); OBS_CT, number of individuals with non-missing data; BETA, effect size of A1 allele; SE, standard error of A1 allele; T_STAT, t-statistic; P, p-value of association between A1 allele and serum urate levels.

• Supplementary file 2. Independent GWAS hits for IGF-1. CHROM, chromosome number; POS, variant position (hg19); ID, variant identifier; REF, reference genome sequence allele; A1, alternative allele; A1_CT, number of A1 alleles; ALLELE_CT, total alleles; A1_FREQ, frequency of A1 allele; MACH_R2, estimated imputation accuracy (INFO); OBS_CT, number of individuals with non-missing data; BETA, effect size of A1 allele; SE, standard error of A1 allele; T_STAT, t-statistic; P, p-value of association between A1 allele and serum IGF-1 levels.

• Supplementary file 3. Independent GWAS hits for testosterone in males. CHROM, chromosome number; POS, variant position (hg19); ID, variant identifier; REF, reference genome sequence allele; A1, alternative allele; A1_CT, number of A1 alleles; ALLELE_CT, total alleles; A1_FREQ, frequency of A1 allele; MACH_R2, estimated imputation accuracy (INFO); OBS_CT, number of individuals with non-missing data; BETA, effect size of A1 allele; SE, standard error of A1 allele; T_STAT, t-statistic; P, p-value of association between A1 allele and serum testosterone levels in males.

• Supplementary file 4. Independent GWAS hits for testosterone in females. CHROM, chromosome number; POS, variant position (hg19); ID, variant identifier; REF, reference genome sequence allele; A1, alternative allele; A1_CT, number of A1 alleles; ALLELE_CT, total alleles; A1_FREQ, frequency of A1 allele; MACH_R2, estimated imputation accuracy (INFO); OBS_CT, number of individuals with non-missing data; BETA, effect size of A1 allele; SE, standard error of A1 allele; T_STAT, t-statistic; P, p-value of association between A1 allele and serum testosterone levels in females.

• Supplementary file 5. Phenotype-level correlations between luteinizing hormone (LH) and testosterone in females and males. Magnitude of correlation and sample sizes are both higher using the XM0lv luteinizing hormone code, but results are consistent across codes.

• Supplementary file 6. Female-specific association with circulating testosterone levels at the FSHB locus. We observe an association between the previously discovered rs11031006 (*Ruth et al., 2016*; *Laisk et al., 2018*) and serum testosterone levels in females. This association was reproduced in the non-British White individuals in UK Biobank. All effects are at rs11031006 with respect to dosage of the A allele.

• Supplementary file 7. Pathways representing core genes for serum urate biology. Pathway, which class of genes; Gene name, name of gene included in the given pathway.

• Supplementary file 8. Pathways representing core genes for serum IGF-1 biology. Pathway, which class of genes; Gene name, name of gene included in the given pathway.

• Supplementary file 9. Pathways representing core genes for serum testosterone biology. Pathway, which class of genes; Gene name, name of gene included in the given pathway.

• Supplementary file 10. SNP heritabilities and level of population stratification as estimated by LD Score regression (*Finucane et al., 2015*) using the full set of baseline and cell-type-specific annotations for each biomarker trait, with height as a baseline. The lower estimates of LD Score regression SNP-based heritability relative to HESS are expected (*Shi et al., 2016*). Best $h^2$ intercept refers to the intercept of the inflation for the best-fitting simulation results in the $h^2$-derived causal SNP estimates (Materials and methods).

• Supplementary file 11. Estimates of SNP heritability and fraction of causal variants from GENESIS (*Zhang et al., 2018*). K, number of mixture components used in the fit of effect sizes. Half sample, 50% downsample of individuals in GWAS to mimic the sex-specific traits. * Failed to converge and terminated after a single iteration.

- Supplementary file 12. Phenotype-level correlations between luteinizing hormone (LH) and testosterone in females and males. Magnitude of correlation and sample sizes are both higher using the XM0lv luteinizing hormone code, but results are consistent across codes.

- Transparent reporting form

## Data availability

Full raw summary statistics and relevant processed data tables are available on Figshare (https://doi.org/10.6084/m9.figshare.c.5304500.v1), or the lab website (http://web.stanford.edu/group/pritchardlab/dataArchive.html, direct link to google drive https://drive.google.com/drive/u/3/folders/10hCG_Wz8f25E6_sxw6sB8vDtS2OWUW9E).

The following dataset was generated:

| Author(s) | Year | Dataset title | Dataset URL | Database and Identifier |
|---|---|---|---|---|
| Naqvi S | 2021 | Supplementary Data for Sinnott-Armstrong and Naqvi | https://doi.org/10.6084/m9.figshare.c.5304500.v1 | figshare, 10.6084/m9.figshare.c.5304500.v1 |

The following previously published datasets were used:

| Author(s) | Year | Dataset title | Dataset URL | Database and Identifier |
|---|---|---|---|---|
| Mesirov J, Tamayo P, Castanza A, Eby D, Medetgul-Ernar K, Niklason J, Reich M, Subramanian A, Thorvaldsdóttir H, Wenzel A, Xu X | 2019 | MSigDB | http://software.broadinstitute.org/gsea/msigdb/ | GSEA, msigdb |

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
