## [Decision Letter]

Thank you for submitting your article "GWAS of three molecular traits highlights core genes and pathways alongside a highly polygenic background" for consideration by *eLife*. Your article has been reviewed by four peer reviewers, one of whom is a member of our Board of Reviewing Editors, and the evaluation has been overseen by Patricia Wittkopp as the Senior Editor. The following individuals involved in review of your submission have agreed to reveal their identity: Vincent J Lynch (Reviewer #2); Naomi Wray (Reviewer #3); Aravinda Chakravarti (Reviewer #4).

The reviewers have discussed the reviews with one another and the Reviewing Editor has drafted this decision to help you prepare a revised submission.

Summary:

Pritchard and colleagues use genome-wide association studies (GWAS) to identify genetic variants in the UK Biobank associated with three molecular traits-urate, IGF-1, and testosterone. Elegant and comprehensive analyses clearly demonstrate that the known biology of these traits explain many of the top hits of the GWAS, even when trait biology differs by sex, and that these core signals reside in a sea of polygenic variation.

The authors interpret their results within the framework of the "omnigenetic" model, namely that there is a difference between genetic effects from variants affecting core biological processes directly related to a trait, and a (usually much larger) number of loci that are not directly related to the trait.

1) The main concern is that the omnigenic hypothesis remains vague and untested. The authors do not address a crucial question: if the major hits are direct and core, are the other hits not core and therefore not important to unravel the biology of the trait?

One test of their hypothesis is that the many other significant associations are at genes that do affect the core genes. Their demonstration, at least for urate, is that the remaining non-core gene heritability is largely from the kidney. This is an important piece of the evidence. But they also need to demonstrate that these peripheral genes affect core gene expression.

They state that "genes that are expressed in trait-relevant cell types are referred to as "peripheral" genes, and can matter if they affect the expression of core genes." This is not demonstrated. The problem with the core versus peripheral definition is that it is imprecise. Is genetic variation in a transcription factor that directly regulates a core gene, core or peripheral?

While this concern does not lessen the value of the GWAS they perform, we'd like to see the authors either address this issue, or re-focus the manuscript so that the context for the GWAS is no longer to test the omnigenic hypothesis.

2) The authors should put their findings, and their interpretation, within the broader context of the literature on quantitative traits. As currently written, the manuscript provides a partial view of complex traits and disease. A better referenced Introduction and Discussion, and acknowledgement that the omnigenic model is consistent with long-published conceptualisations is needed.

For example, they conclude "these vignettes help to illustrate why many diseases are extraordinarily polygenic, as they are usually impacted by multiple biological processes that, like those considered here, are themselves highly polygenic." It needs to be pointed out that this conclusion is consistent with the thinking of the last 50 years.

Some acknowledgement needs to be made that many authors have reported that core biology can be recovered through top hits of a GWAS.

There is a long history of considering molecular phenotypes as endophenotypes or "intermediate" phenotypes. That literatures should be cited and the authors' results related to it. Without additional referencing Figure 9 is presented to the reader as if no others have conceptualised common disease as the endpoint of many contributing polygenic traits ("The point here is that when multiple risk factors-each of which is polygenic-contribute to any given disease, the disease endpoint absorbs the polygenic basis for all of the risk factors together."). For example, Figures 1 and 2 of Gottesman and Gould are conceptually very similar to Figure 9 (https://doi.org/10.1176/appi.ajp.160.4.636). The review of milk coagulation traits in 2012 (http://dx.doi.org/ 10.3168/jds.2012-5507) provides a Figure (#5) conceptually similar to Figure 9. There are other examples (e.g., PLoS Genet 6(9): e1001139. doi:10.1371/journal.pgen.1001139).

The first paragraph of the Discussion states "We showed that unlike most disease traits, these three biomolecules have clear enrichment of genome-wide signiﬁcant signals in core genes and pathways." The statement reads as if it were a novel finding, but it is expected that as a DNA-trait relationship gets closer SNP associations will be biologically more obvious (and indeed the terms core and peripheral genes have been used long before the advent of the omnigenic model).

Revisions expected in follow-up work:

1) Authors should address the issue of whether their findings can address key elements the omnigenic hypothesis, or re-focus the manuscript so that the context for the GWAS is no longer so tied to the omnigenic hypothesis.

2) Put their findings, and their interpretation, within the broader context of the literature on quantitative traits.

---

## [Author Response]

Revisions for this paper:1) The main concern is that the omnigenic hypothesis remains vague and untested. The authors do not address a crucial question: if the major hits are direct and core, are the other hits not core and therefore not important to unravel the biology of the trait?

We respectfully disagree that the hypothesis remains vague, and in that regard, point to Liu et al., 2019, which lays out our model in detail, elaborating on the verbal model in Boyle et al., 2017. Liu et al. sought to develop a model to understand why it is that so many variants, spread so widely across the genome, can be responsible for the heritability of a typical complex trait. Very briefly, Liu et al. proposed that most heritability is due to variants with cis-effects on peripheral genes. These in turn perturb the regulation of a smaller class of core genes via trans regulatory networks; genetic effects on traits are mediated via core genes. Thus, peripheral genes are important to unraveling the biology of a trait because they explain the trans-regulatory context within which core genes sit.

That said, we acknowledge that at this time the model remains largely conceptual. There are no traits that are understood in sufficient detail to fully evaluate the model. For most traits the core genes are unknown; further, at this time we do not know detailed trans-regulatory networks in any cell type. Nonetheless, we would argue that there is value in articulating specific models for how genetic variation impacts traits. In population genetics and quantitative genetics there is a long history of theoretical models that preceded the relevant data, and – whether these have turned out to be right or wrong – they have been invaluable in guiding and informing future research questions.

In this paper we elucidate one major component of the model, namely the identities and roles of core genes, for three example traits. For most disease traits there is only very incomplete knowledge of likely core genes; here we use these three molecular traits to (1) Provide examples of core genes, and to show huge enrichment of signal around those, and

(2) To show that in these examples, variation at core genes explains only a small fraction of the heritability.

By doing so, our paper starts to connect the dots for certain key aspects of the model that have been difficult to verify in more complicated traits. Clarifying the nature and roles of core genes in specific examples is an important step, as some commentators, including one of the reviewers of this paper, have argued against the concept of core genes. We acknowledge, however, that our paper does not touch on the network part of the model (as now mentioned in the Introduction) – this remains for future work. In summary: we have reframed the manuscript, including the Introduction, to make clear that this study focuses on specific aspects of the omnigenic model that have not, to date, been rigorously tested with real data, while not yet addressing all components of the model.

The authors do not address a crucial question: if the major hits are direct and core, are the other hits not core and therefore not important to unravel the biology of the trait?

We have added a consideration of this and related issues to the Discussion. In brief, if tens of thousands of variants, acting through at least thousands of genes, have nonzero effects, then presumably most of these are not going to be useful for understanding the biological mechanism of a disease.

One test of their hypothesis is that the many other significant associations are at genes that do affect the core genes. Their demonstration, at least for urate, is that the remaining non-core gene heritability is largely from the kidney. This is an important piece of the evidence. But they also need to demonstrate that these peripheral genes affect core gene expression.

We agree that as an additional step to evaluate the omnigenic model, it will be important to study regulation of core gene expression by delineating cellular regulatory networks. While an experimental demonstration of such regulatory connections is beyond the scope of this paper (as we now note in the Introduction), we have analyzed cis- vs. trans-heritability of expression of the sets of curated core genes for each of the model traits in this paper. As a set, we have found that core genes do not have significantly higher expression cis-heritability than all other genes (Figure 8—figure supplement 14). This is consistent with the idea that much of core gene expression is determined in trans, presumably through regulation by peripheral genes. A more comprehensive analysis of this would require enumerating cis vs. trans heritabilities of genes in a variety of tissues, which would be an immense project outside the scope of the current work.

They state that "genes that are expressed in trait-relevant cell types are referred to as "peripheral" genes, and can matter if they affect the expression of core genes." This is not demonstrated. The problem with the core versus peripheral definition is that it is imprecise. Is genetic variation in a transcription factor that directly regulates a core gene, core or peripheral?

In this study, we have defined core genes in line with the definition from Liu et al., 2019. In that light, a transcription factor that regulates other core genes would be a peripheral gene, or perhaps a master regulator if it simultaneously regulates multiple core genes. However, as defined by Liu et al., we consider signaling receptors, such as the androgen receptor for testosterone, to be core even though they are transcription factors, since they directly receive inputs from outside the cell. Transcription factors that act within a cellular regulatory network , rather than directly receiving inputs from outside the cell, would be considered peripheral. Nevertheless, we admit that this is a simplified, conceptual model, such that not all genes will fit neatly into these definitions. We have made these points more clearly in the Introduction.

While this concern does not lessen the value of the GWAS they perform, we'd like to see the authors either address this issue, or re-focus the manuscript so that the context for the GWAS is no longer to test the omnigenic hypothesis.

As noted above, the contribution of this paper to the omnigenic model is in: (1) illustrating the role of core genes for specific traits; (2) showing that these contribute only a small fraction of the heritability; and (3) showing that even these seemingly simpler traits are affected by on the order of 10,000 genes.

2) The authors should put their findings, and their interpretation, within the broader context of the literature on quantitative traits. As currently written, the manuscript provides a partial view of complex traits and disease. A better referenced Introduction and Discussion, and acknowledgement that the omnigenic model is consistent with long-published conceptualisations is needed.

We disagree with the vague claim that “the omnigenic model is consistent with long-published conceptualisations” as well as the implication below that it is “consistent with the thinking of the last 50 years” and believe they represent a misunderstanding of the central question addressed in this manuscript.

Prior to ~2006, researchers in human genetics had almost no idea how many loci might underlie variation in complex traits, and many expected that this might be on the order of tens of loci, and that some loci would have large effect sizes (see references in our Introduction). Only in the last few years has it become clear how many causal variants there are, and that most of the genome contributes to heritability. While we appreciate that theoretical models in quantitative genetics had considered the possibility that many loci might contribute to variation in a trait, there was no empirical evidence that they applied to humans. Most importantly, those studies were not focused on – or even interested in – the mechanisms that led to polygenic inheritance.

In that regard, these new findings pose a new question that would not have seemed relevant prior to the GWAS era: From a mechanistic point of view, how should we interpret the observation that so many variants (and by extension so many genes) contribute to any given trait, and that the lead variants contribute such a small fraction of heritability? The central element of our model rests on the roles of cis and trans-acting expression QTLs. The observation that most expression variance is due to many small trans effects has emerged relatively recently – roughly in the last dozen years – and those results have, arguably, been underappreciated in the field. Thus, beyond the much more generic claim that complex trait variation may be highly polygenic, we’re not aware that anyone else has formulated the specific model previously articulated by Boyle et al., 2017, and Liu et al., 2019.

For example, they conclude "these vignettes help to illustrate why many diseases are extraordinarily polygenic, as they are usually impacted by multiple biological processes that, like those considered here, are themselves highly polygenic." It needs to be pointed out that this conclusion is consistent with the thinking of the last 50 years.

The specific comment here refers to a section in the Discussion noting that when there are multiple biological processes contributing to a disease these can further increase polygenicity in addition to the main themes that we have discussed. This is not really part of the main omnigenic model, which proposes a key role for intracellular networks. We mentioned the multiple-processes point as it presumably also contributes to the polygenicity of many disease endpoints; that said it seems less likely to explain simpler traits such as those studied here, especially urate where the heritability comes almost entirely from the kidneys. Thus, this study provides empirical evidence for one model of why disease polygenicity is so high.

Given that the presence of this section in the Discussion seems to have been distracting, we removed Figure 9 and rewrote the corresponding paragraph to make clearer that we think of this as a complementary model rather than part of the same model.

Some acknowledgement needs to be made that many authors have reported that core biology can be recovered through top hits of a GWAS.

We had reviewed the past literature on core genes in our previous papers on this topic (Boyle et al., 2017, Liu et al., 2019). It seems outside scope to cover that at great length again, but we now point readers to some of this literature with the following:

“We do now know various examples of core genes or master regulators for specific traits (e.g., Sekar et al., 2016, Small et al., 2011, Small et al., 2018), but there are few traits where we understand the roles of more than a few of the lead genes. Among the clearest examples in which a whole suite of core genes have been identified are for plasma lipid levels (e.g., Liu et al., 2017, Lu et al., 2017, Hoffmann et al., 2018), reviewed by Dron et al., 2016, Liu et al., 2019; and for inflammatory bowel disease (de Lange et al., 2017).”.

There is a long history of considering molecular phenotypes as endophenotypes or "intermediate" phenotypes. That literatures should be cited and the authors' results related to it. Without additional referencing Figure 9 is presented to the reader as if no others have conceptualised common disease as the endpoint of many contributing polygenic traits ("The point here is that when multiple risk factors-each of which is polygenic-contribute to any given disease, the disease endpoint absorbs the polygenic basis for all of the risk factors together."). For example, Figures 1 and 2 of Gottesman and Gould are conceptually very similar to Figure 9 (https://doi.org/10.1176/appi.ajp.160.4.636). The review of milk coagulation traits in 2012 (http://dx.doi.org/ 10.3168/jds.2012-5507) provides a Figure (#5) conceptually similar to Figure 9. There are other examples (e.g., PLoS Genet 6(9): e1001139. doi:10.1371/journal.pgen.1001139).

Thank you for these additional citations. We note that our previous version did in fact reference previous papers in this context, including Udler, 2019, who had a similar figure, and Turkheimer, 2000, who has written extensively on this point in the context of behavioral genetics. That said, as noted above, we have removed Figure 9 as this was somewhat extraneous to our main points, shortened the corresponding section in the Discussion, and added these additional references.

The first paragraph of the Discussion states "We showed that unlike most disease traits, these three biomolecules have clear enrichment of genome-wide signiﬁcant signals in core genes and pathways." The statement reads as if it were a novel finding, but it is expected that as a DNA-trait relationship gets closer SNP associations will be biologically more obvious (and indeed the terms core and peripheral genes have been used long before the advent of the omnigenic model).

Here, use of the word “expected” suggests that perhaps the reviewers are referring to their expectations rather than specific prior data analyses that demonstrate this point.

With regard to the three biomolecules considered here, prior GWAS analysis showed mixed evidence for enrichment in the relevant pathways. While the urate transporters were well-known to be enriched for GWAS hits, enrichment in the urate synthesis pathway was not demonstrated, and no such pathway enrichments were demonstrated for IGF-1. Similarly, Ruth et al., 2020, did not focus on testosterone biology as the focus was mainly on sex differences in genetic effects and their relationship to cardiometabolic traits, and previous papers (e.g. Ohlsson et al., 2011, Ruth et al., 2015, and Prescott et al., 2012) focus almost exclusively on the strong association with SHBG variants in males and have no genome-wide significant associations in females (though Prescott et al., 2012 reports a single CYP4A1 subthreshold association in passing as well).

Revisions expected in follow-up work:1) Authors should address the issue of whether their findings can address key elements the omnigenic hypothesis, or re-focus the manuscript so that the context for the GWAS is no longer so tied to the omnigenic hypothesis.

We have clarified that in this paper we elucidate one major component of the model, namely the identities and roles of core genes. For most disease traits there is only very incomplete knowledge of likely core genes; here we use these three molecular traits to (1) Provide examples of core genes, and to show huge enrichment of signal around those, and

(2) To show that in these examples, core genes explain only a small fraction of the heritability. We further clarified that the paper does not address the network component of the model.

(2) Put their findings, and their interpretation, within the broader context of the literature on quantitative traits.

We have expanded the Introduction and Discussion to provide more context on the history of studies of complex traits.